# Modelling 2050 Water Retention Scenarios for Irrigated and Non-Irrigated Crops for Adaptation to Climate Change Using the SWAT Model: The Case of the Bystra Catchment, Poland

**Damian Badora** *[ID], **Rafał Wawer** *[ID] **and Aleksandra Król-Badziak**

The Institute of Soil Science and Plant Cultivation—State Research Institute, ul. Czartoryskich 8, 24-100 Pulawy, Poland
* Correspondence: dbadora@iung.pulawy.pl (D.B.); huwer@iung.pulawy.pl (R.W.)

**Abstract:** The paper presents the estimated changes in the soil water content, the total runoff, the sediment yield and the actual evapotranspiration for the small Bystra catchment in the east of Poland. The findings are based on the results of three simulations covering the years of 2041–2050. The simulations were based on a calibrated and validated SWAT model (2010–2017). The first variant covers just the climate change and the existing structure of soil cultivation for the three regional climate models supported by the EC-EARTH global climate model in the emission scenarios RCP4.5 and RCP8.5. Variants two and three are based on the first variant in terms of the changing climate. The second variant, however, involves placing a pond in each farm in the catchment, while the third variant involves designing huge reservoirs as a result of land consolidation. Variants two and three occur in five adaptation scenarios each. The first adaptation scenario (V2.1 and V3.1) involves only increasing the number of ponds on the farm or increasing the number of reservoirs for non-irrigated arable land crops, i.e., WWHT (winter cereals), BARL (spring cereals), CANP (rapeseed) and CRDY (other crops). The second adaptation scenario (V2.2 and V3.2) involves growing vegetables without irrigation (instead of cereals). The third adaptation scenario (V2.3 and V3.3) involves growing vegetables with irrigation (instead of cereals). The fourth adaptation scenario (V2.4 and V3.4) involves partial cultivation of vegetables and cereals. The fifth adaptation scenario (V2.5 and V3.5) involves partial cultivation of orchards and cereals. The adaptation scenarios of the irrigation of vegetables from deep water-bearing layers (second variant) or reservoirs (third variant) contribute to the increase in water content in the soil, especially in summer, in comparison with the adaptation scenarios for vegetable cultivation without irrigation. What is more, the actual evapotranspiration was higher in the adaptation scenarios involving irrigation than in scenarios without irrigation. It is known that the changes in water content in soil and the intensification of water erosion are gravely affected by modifications in crops and soil cultivation. A change from cereal cultivation to irrigated vegetable cultivation or orchards increased the water content in the soil in most climatic projections. However, the increase in the number of ponds in the second variant had little impact on the soil water content, actual evapotranspiration and overall runoff, while the erosion loss decreased. With the lower precipitation levels in the years 2041–2050 relative to 2010–2017, as presented in the emissive scenario RCP 4.5, the soil water content decreases by up to 14% for most variants. Total runoff for most variants will also be lower by 4–35%. The percentage change in sediment yield will fluctuate between −86% and 116%. On the other hand, the actual evapotranspiration for most variants will be higher. With higher precipitation levels in the years 2041–2050 relative to 2010–2017, as presented in the emissive scenario RCP 8.5, the soil water content changes slightly from −7% to +3%. Total runoff for most variants will also be higher by as much as 43%. Sediment yield for most scenarios may increase by 226%. The actual evapotranspiration for most variants will also be higher. Irrigation variants tend to increase soil available water while increasing evapotranspiration and total outflow in the catchment as compared to non-irrigated LULC. The largest increase in the soil water content is observed in most irrigation variants for RCP 4.5 (annual average 316–319 mm) (V2.3-V2.5, V3.2, and V3.3) and RCP 8.5 (annual average 326–327 mm) (V2.3-V2.5 and V3.3) as compared to V1 (BaU) (315 mm–RCP 4.5 and 324 mm–RCP 8.5) for the years 2041–2050. On the other hand, the lowest increase in

soil water content is observed in the V3.5 variant, with an annual average of 292 mm for RCP 4.5 and an annual average of 311 mm for RCP 8.5. Thus, for future climate change scenarios, irrigation with water reservoirs (ponds and storage reservoirs) should be considered. The study proves the rationale behind building ponds in small catchments in order to increase water resources in a landscape and also to counteract adverse effects of climate changes, i.e., sediment outflow and surface water erosion.

**Keywords:** SWAT; SWAT-CUP; climate change; adaptation scenarios; small retention; soil water content; total runoff; sediment yield; actual evapotranspiration; crop change; irrigation; ponds; reservoirs

## 1. Introduction

Water resources of a catchment depend, among others, on the amount of precipitation, evapotranspiration, temperature, as well as on the properties of soils (water storage capacity, infiltration rate, texture and structure), management and vegetation [1,2]. One of the important elements of the hydrological cycle is the water content of the soil [3], which influences many processes occurring in the catchment (e.g., erosion) and vegetation. The primary source of soil water is precipitation through infiltration and surface runoff [4]. Temperature, on the other hand, affects the evapotranspiration process [5].

Many hydrologic models are currently used in research. One such model is the Soil Water Assessment Tool (SWAT) model developed by the U.S. Department of Agriculture (USDA), which is widely used by researchers [6,7]. The SWAT model was selected for this study because of its ability to predict the effects of land management practices on hydrology and water quality in a watershed. A considerable number of studies using the Soil and Water Assessment Tool (SWAT) model for estimating the content of water in soil relate to large catchments such as the Vistula catchment [8–10]. However, there are only a few studies concerning small catchments in Poland. These include publications on the Barycz and Górna Narew catchment [11] and the Bystra River catchment [12], which describe research on the soil water content, total runoff, sediment yield and actual evapotranspiration using climate scenarios.

The need to study small catchments covering up to several hundred square kilometers using suitable soil parameters (e.g., retention) is justified by the possibility of modeling the catchment environment based on high resolution input data, which facilitates the interpretation of results. The following parameters of soil retention: available water capacity (AWC), wilting point (WP) and field capacity (FC) were designated for soil types occurring in Poland by a team of researchers at IUNG-PIB [2]. The SWAT model enables simulating the effectiveness of the adaptation of farming and spatial economy to the changing climate, thus offering a choice of optimal sustainable adaptation strategies for a given area. The Bystra catchment has been selected for this research for several of reasons, including its large share of farming land, varied landforms, prevalence of loessal soil and, last but not least, the accessibility of data from the watercourse hydrology monitoring system.

The change in climate in the decades to come is the subject of much research today. The associated unpredictable weather phenomena give rise to many concerns related to the environmental, social and economic risks that may arise. Climate change will also affect agriculture in Poland [13,14].

Previous studies have evidenced an increase in air temperature in recent decades, which was also contributed to the increase in potential evapotranspiration and increased variability of this indicator, especially visible in recent years (2011–2020) [12,15]. The increase in evapotranspiration, especially during the growing season in recent decades, has also been the subject of other studies [16]. In addition, changes in climate have been observed in Poland, resulting from warming, as well as changes in precipitation and a few weather extremes [17,18].

According to the Intergovernmental Panel on Climate Change (IPCC), the Earth's average surface temperature will reach 1.5 degrees Celsius above pre-industrial levels in

the coming decades [19]. In 2007, it was reported that droughts could increase 10-fold over the next decades [20]. Droughts observed during that time suggest that projections using climate change models seem to reflect the changing climate in Poland quite well [21]. In recent years in Poland, there has been a tendency to pay more attention to implementing various adaptation measures, e.g., in agriculture. Beforehand, the focus was on alleviating climate change. State announcements frequently presented climate forecasts, susceptibility to these changes and the effects of climate change [22].

The increase in temperature and precipitation, and thus also evapotranspiration in the coming decades, will also apply to the Vistula river basin [16] and Europe [23,24].

Therefore, it is necessary to find solutions mitigating the negative effects of climate change [25], e.g., the appearance of extreme weather phenomena including droughts and sudden floods [15,26,27] in the Bystra catchment over the next several decades.

In order to properly assess the solutions presented in this paper, a business as usual (BaU) scenario was investigated, which shows the changes in the water balance of the catchment caused solely by climate change, with the anthropogenic factors being unaltered. The BaU scenario for 2050 is based on the SWAT model and is calibrated and validated on archival data. The article aims to analyze three variants of small retention with respective adaptation scenarios for farming practices:

**V1**: 'Business as Usual'. Same retention, same land use and land cover (LULC), using 2050 climate predictions;

**V2**: (V2.1, V2.2, V2.3, V2.4 and V2.5). Small retention reservoir. Small pond on every farm, five LULC and irrigation scenarios, using 2050 climate predictions;

**V3:** (V3.1, V3.2, V3.3, V3.4 and V3.5). Big retention. Big ponds achieved via land improvements, five LULC and irrigation scenarios, using 2050 climate predictions. The scheme covers the years 2041–2050 with regard to three regional climate models (RCM) powered by a general circulation model (GCM) from RCP 4.5 and RCP 8.5 emission scenarios.

The soil water content for the Bystra catchment in the SWAT model for the years 2010–2017 was compared to six climate projections (three GCMs/RCMs × two RCPs) for the years 2041–2050. The findings are shown in the Results section of a paper concerning the impact of farming adaptation practices on the soil water content in the future climate for the Bystra catchment area [28]. The lower amount of water in the soil in most seasons (2041–2050) as compared to the SWAT model (2010–2017) calls for a consideration of the introduction of variants into the model by using adaptation scenarios specifically for the sake of this paper, the aim of such scenarios is to maintain or increase the soil water content in the coming decades.

This paper presents a comparison of the results of soil water content (1.5 m profile), actual evapotranspiration, total runoff and sediment yield for three variants (the 2nd and 3rd variants each contain five adaptation scenarios) that were derived from simulations of the calibrated and validated SWAT model [12] for three regional climate models derived from the EC-EARTH global climate model for 2041–2050 (variant 1).

The three variants include climate change only, a pond design for each farm in the catchment and a reservoir design. Variants 2 and 3 each include five adaptation scenarios.

The first adaptation scenario involves just the increase in the number of ponds on the farm or the increase in the number of reservoirs for non-irrigated arable land crops, i.e., WWHT (winter cereals), BARL (spring cereals), CANP (rapeseed) and CRDY (other crops). The second scenario involves growing vegetables without irrigation, instead of cereals. The third scenario involves growing vegetables with irrigation, instead of cereals. The fourth scenario involves partial cultivation of vegetables and cereals. The fifth scenario involves partial cultivation of orchards and cereals.

## 2. Material and Methods

### 2.1. Characterization of the Study Area

The Bystra River, located in the Lubelskie Province, is a right bank tributary of the Vistula, and is 33 km long (Figure 1). According to the digital elevation model (DEM)



generated by the SWAT program, the lowest point of the basin is 126 m above sea level, the highest point is 246 m above sea level and the area of the basin is 296.6 km² [12].

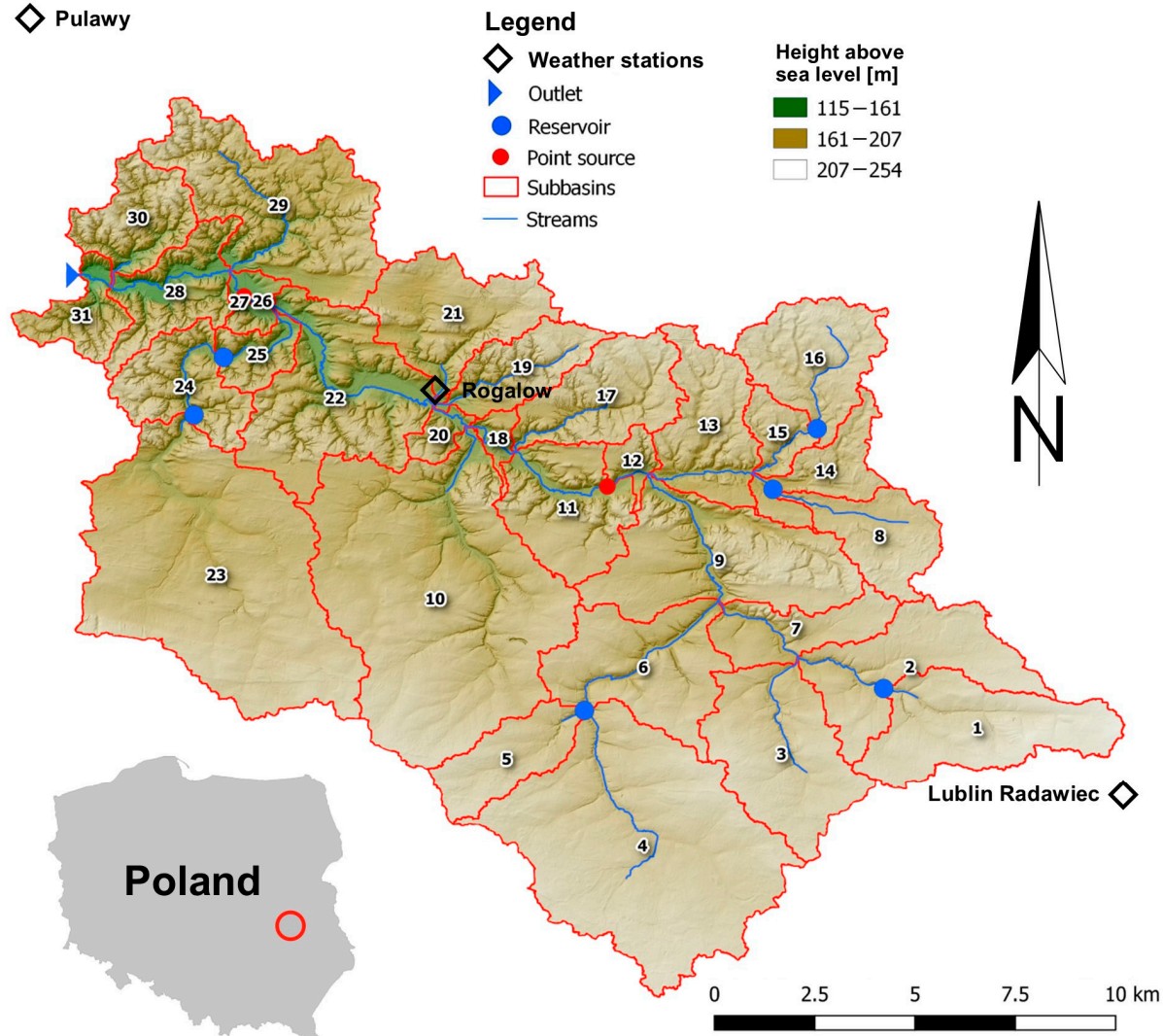

**Figure 1.** The catchment area of the Bystra River [28].

The area of Bystra in the Lublin Upland [29–31] is characterized by a highly diverse terrain relief, consisting of many valleys with a constant or episodic tributary, especially in the north-western part. The main valley of the Bystra is 35 km long, which at the mouth of the Vistula River cuts up to 35 m into marl and bedrock [32–34]. Such relief of the catchment area poses a significant threat in terms of moderate and very strong water surface erosion [35]. Geologically, most of the catchment area is composed of deep loess of up to 20 m [36].

The study area consists of mostly podzolic and lessivage soils (49%), as well as cambisoils (47%) [12]. The soils in the catchment area are composed mainly of loess (73%) [37–40] and silt (18%) [12].

Agricultural land (78%) and forests (16%) dominate the Bystra catchment area. The agricultural land consists mostly of non-irrigated arable land (52%), fruit trees and plantations (11%), complex cultivation patters (9%) and pastures (6%) [12].

### 2.2. Description of SWAT Model and SUFI-2 Model

The water balance is defined by the quantitative water circulation in a catchment at a given time, allowing the estimation of permanent and temporary water resources occurring in the area [41]. Basin modelling takes place in two phases: land [41] and routing [42].

One of the basic formulas used in SWAT modelling is the water balance equation:

$$SW_t = SW_0 + \sum_{i=1}^{t} (P_d - SURQ - E - w_{seep} - GWQ) \tag{1}$$

where $SW_t$ is the final water content of the soil (mm); $SW_0$ is the initial water content of the soil (mm); $t$ is the time in days; $P_d$ is precipitation (mm); $SURQ$ is surface runoff (mm); $E$ is evapotranspiration (mm); $w_{seep}$ is the amount of water entering the vadose zone from the soil profile (mm); and $GWQ$ is the groundwater flow (mm) [7].

The SWAT model was calibrated and validated using the SUFI-2 algorithm to achieve an accurate projection of the catchment parameters [43–45]. SUFI-2 is an adequate algorithm for small catchments [46,47].

### 2.3. Data Used in the SWAT Model and SWAT-CUP Program

An overview of the input data used in the SWAT model and the SWAT-CUP program is provided in Table 1.

**Table 1.** Input data used in SWAT model and SUFI-2 algorithm [12].

| Data Type | Data Description, Processing and Usage | Scale/Resolution | Source |
|---|---|---|---|
| Topography | DEM | 5 m | [48] |
| Soils | Shapefile | 1:25,000 and 1:100,000 | [49,50] |
| Land Use | Shapefile | 1:100,000 | [51] |
| Hydrographic | Shapefile | 1:50,000 | [52] |
| Open Street Map | Shapefile | - | [53] |
| Geological | Shapefile | - | [36] |
| Orthophotomap | WMS | High resolution | [54] |
| Weather | Precipitation (mm), temperature (°C), wind speed (m/s), humidity and solar total radiation $(MJ/m^2)$ | Daily | Statutory research of IUNG-BIP and other [55] |
| Sewage treatment plants | Average daily water loading $(m^3/day)$ | Daily | [56] |
| Out flow | Calibration and validation $(m^3/month)$ | Monthly | Statutory research of IUNG-BIP |

### 2.4. SWAT Model and SUFI-2 Algorithm

The chosen model has 31 partial catchments (Figure 1), and the soil data were collected from the statutory data of IUNG-PIB [12]. The AWC and WP values were obtained from a previous water balance study in the research basin [2].

The CORINE Land Cover map has been enriched with additional vectorization of land cover and land use of the Bystra catchment area in order to increase the resolution of land use using an orthophotomap and open street map data.

The research area was divided according to the decline in the area in the following ranges: 0–6%, 6–10%, 10–18%, 18–27% and >27%. The above slope ranges are derived from the PWER and AWER indicators [57] for the risk of soil erosion, which remain the standard in visualization of land relief in Poland.

In the study area, 484 hydrologic response unit (HRU) areas were established. During the creation of HRU areas, the CRDY arable lands were additionally defined, from which WWHT winter crops (43%), BARL spring crops (31%), CANP rape (14%) and CRDY (12%) were distinguished, based on the publication of agricultural use data in the Lubelskie Voivodeship in 2019 [58]. APPL apple orchards were isolated from ORCD fruit orchards [58]. The forests were divided into coniferous FRSE forests (49%), deciduous FRSD forests (13%) and mixed FRST forests (38%) according to the data obtained from the Regional Directorate of State Forests in Lublin [12,59].

### 2.5. Meteorogical Data

The meteorological data for the years 2005–2017 input in the SWAT model we obtained from three weather stations in Puławy, Rogalów and Lublin Radawiec [12]. The SWAT model has also been supplemented and corrected for some parameters related to the point of discharge of wastewater and for the parameters of planned non-irrigated arable land management operations. The current value of $CO_2$ concentration was also entered into the model.

Then, a simulation of the SWAT model for 2010–2017 was carried out at monthly intervals with a five-year model start-up period.

### 2.6. SWAT CUP Calibration and Validation Results

At the next stage, the model was calibrated and validated using the SUFI-2 algorithm to obtain a more realistic picture, making use of the average monthly observations of the outflow at the Bystra estuary for the years 2010–2014 (calibration) and 2015–2017 (validation) [12]. For NSE, the following values were obtained: the calibration was 0.58 and the validation was 0.70. However, for $R^2$, the values obtained were a calibration of 0.60 and a validation of 0.71 [12]. Both NSE and $R^2$ values fall within satisfactory ranges [43,60,61].

In order to verify the adequacy of the calibrated and validated SWAT model, we compared the values of potential evapotranspiration with the results obtained from the statutory findings [62]. In both studies, the values of potential evapotranspiration correspond.

### 2.7. Climate Change Scenarios

The daily climate data grid used in this study has been prepared and tested in a publication on the calibration and validation of the SWAT model for the current and future water balance of the Bystra catchment [12]. The three climate projections driven by two RCP (Representative Concentration Pathways) were selected for further study in terms of temperature and precipitation changes. They reflect a range of possible climate variations, and thus cover uncertainties in the possible climate change projections. Most of the data were obtained at 0.11° spatial resolution from the EURO-CORDEX database for 1951–2050 (available through Earth System Grid Federation at https://esgf-data.dkrz.de/search/cordex-dkrz for Europe, accessed on 15 June 2021) [23,63].

The minimum and maximum daily air temperature, daily precipitation, solar radiation, daily mean wind speed and relative humidity in the climate scenario are based on regional climate models (RCMs) (RACMO22E, HIRHAM5 and RCA4) for two representative concentration pathways (RCP) (RCP 4.5 and RCP 8.5). The RCM models were driven by one general circulation model (GCM), EC-EARTH. Six climate projections (three RCM × two RCP) were used for this study. The RCP corresponds to the radiative forcing values in 2100 with respect to the pre-industrial values. These are +4.5 Wm$^{-2}$ and +8.5 Wm$^{-2}$ [23,64,65] (Table 2). Table 2 also shows the limits of the changes in the characteristics of the selected models for the 2021–2050 period relative to the 1971–2000 baseline period.

**Table 2.** Description of GCM/RCM simulations divided depending on radiative forcing. A comparison of temperature and precipitation changes in 2021–2050 in the GCM/RCM simulations for RCP 4.5 and RCP 8.5 for the base period 1971–2000 [28].

| Models | Scenario Assumptions | | | | Radiative Forcing | |
|---|---|---|---|---|---|---|
| | Change in Average Annual Air Temperature | | Change in Average Annual Precipitation | | +4.5 Wm$^{-2}$ | +8.5 Wm$^{-2}$ |
| GCM/RCM Simulation | RCP 4.5 | RCP 8.5 | RCP 4.5 | RCP 8.5 | RCP 4.5 | RCP 8.5 |
| EC-EARTH/RACMO22E | +1.5 °C | +1.8 °C | +15% | +6% | RCP 4.5.1 | RCP 8.5.1 |
| EC-EARTH/HIRHAM5 | +1.6 °C | +1.9 °C | +12% | +5% | RCP 4.5.2 | RCP 8.5.2 |
| EC-EARTH/RCA4 | +1.6 °C | +2.2 °C | +15% | +11% | RCP 4.5.3 | RCP 8.5.3 |

Data for the climate projections used in the SWAT model were extracted from grid cells that correspond to the location of meteorological stations. Air temperature and precipitation data were bias-corrected by the SMHI (Swedish Meteorological and Hydrological Institute) using the distribution-based scaling (DBS) method [47] and regional reanalysis MESAN (mesoscale analysis) for the 1989–2010 dataset [66]. These data were taken in a rotated polar grid. Therefore, bilinear interpolation was applied, and the datasets were transformed to a common latitude and longitude grid using CDO (Climate Data Operators) software [67].

For the analysis of climate projections (RCP 4.5.1, RCP 8.5.1, RCP 4.5.2, RCP 8.5.2, RCP 4.5.3 and RCP 8.5.3) (Table 2), one iteration in SWAT-CUP was used, which included the best calibration parameters for 2010–2015 [12]. The RCP 4.5 and RCP 8.5 scenarios were modified using $CO_2$ concentrations for 2041–2050, as developed by the Potsdam Institute for Climate Impact Research [68,69].

For the purpose of this paper, the soil water content data for 2041–2050 were used for different climate projections, which were presented in a paper on projected soil water content in 2050 using adaptation scenarios that included changes in land use and agricultural practices [28].

## 3. Results

### 3.1. Variants 1–3 and Adaptation Scenarios 1–5

This article used the SWAT model for the Bystra catchment for years 2010–2017, which was developed in a study on the hydrology of the Bystra catchment in climate projections for the years 2020–2050 for the climate change scenarios RCP 4.5 and RCP 8.5 [12].

For the purpose of this paper, three water retention variants were developed. In addition, five adaptation scenarios each were developed for variant 2 and variant 3. The purpose of the adaptation scenarios was to test the feasibility of increasing soil water content in 2041–2050.

Variant 1 only covers climate change. A detailed description of the water balance and results (total runoff and actual evapotranspiration) have been presented in a paper on modelling the upland hydrology of the Bystra catchment in 2050 using RCP 4.5 and RCP 8.5 emission scenario projections [12], and in a paper on predicted soil water content in 2050 using adaptation scenarios that include changes in land use and agricultural practices [28].

In the second variant, 1505 ponds with dimensions of 50 × 100 m (5000 m$^2$) and a depth of 3 m were designed for the catchment area of the Bystra River. The tanks were designed on the basis of the Polish construction law, according to which, a pond with an area greater than 1000 m$^2$ (but not exceeding 5000 m$^2$) and a depth of no more than 3 m can be built in Poland without going through long procedures of building and legal-water permits. It was also assumed that one such pond would be constructed for every two farms. In the Lublin Province, the average area of an individual farm above 1 ha of agricultural land in 2015 was 8 ha [58]. The total area of ponds in the catchment is 752 ha, which together with existing reservoirs and watercourses represents 2.7% of the catchment area. For the second variant, the soil and arable land were modified to include the projected ponds.

Variant 3 assumes the creation or modification of retention reservoirs in the Bystra catchment (Figure 2).

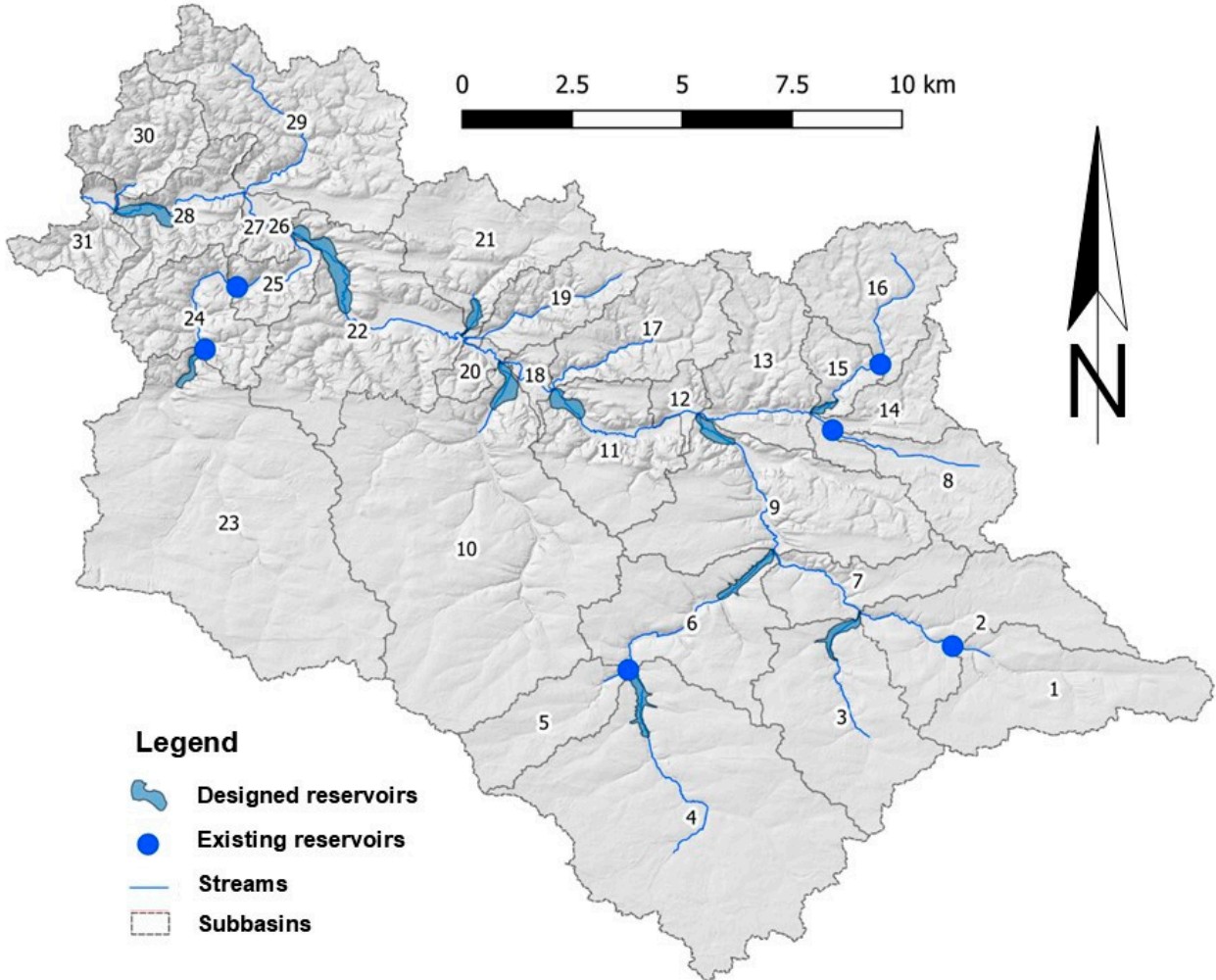

**Figure 2.** Bystra catchment area with modified or designed reservoirs for adaptation scenarios 1–5 for variant 3 (author's own study).

In the designed reservoirs, the area and water capacity were calculated by taking into account the topography (Table 3), both for the emergency area and volume, as well as for the main surface and volume. The total area of the retention reservoirs is 321 ha, which, together with the existing reservoirs and watercourses, constitutes 1.3% of the catchment area.

For variants 2 and 3, five adaptation scenarios were created. The first adaptation scenario involved only increasing the number of ponds on the farm (variant 2) or increasing the number of reservoirs (variant 3) for non-irrigated arable land CRDY, which represents 52% of the Bystra catchment area [12].

In the SWAT model, crops were separated from non-irrigated arable land, CRDY: WWHT (winter cereals, 43%), BARL (spring cereals, 31%), CANP (rapeseed, 14%) and CRDY (other crops, 12%) [12].

In the second adaptation scenario, all crops from scenario one were modified: WWHT, BARL, CANP and CRDY were modified to vegetable crops, i.e., POTA (potatoes, 43%), SGBT (beets, 31%), CRRT (carrots, 14%) and CRDY (12%). The vegetables in this scenario do not have automatic irrigation. However, automatic irrigation was applied to the above-mentioned vegetables in the third scenario. Automatic irrigation is present when the water

content of the soil decreases [6]. Irrigation water will be drawn from a deep aquifer for variant 2 and from created or modified retention reservoirs for variant 3.

In the fourth adaptation scenario, BARL and CANP crops were modified to irrigated vegetable crops, i.e., beet (SGBT) and carrot (CRRT). The same automatic irrigation as in the third adaptation scenario was applied. Crops WWHT and CRDY were left without irrigation, as in scenario one.

In the fifth adaptation scenario, BARL and CANP crops were modified to an irrigated ORCD (orchard). The same automatic irrigation as in the third adaptation scenario was used. Crops WWHT and CRDY were left without irrigation, as in scenario one.

**Table 3.** Areas and volumes of modified or designed reservoirs for adaptation scenarios 1–5 in variant 3 in the Bystra river basin (author's own study).

| Subcatchment | Emergency Area (ha) | Emergency Volume (m$^3$ 10$^4$) | Principal Area (ha) | Principal Volume (m$^3$ 10$^4$) | Above Mean Sea Level (AMSL) |
|---|---|---|---|---|---|
| 1 | 4 | 8 | 4 | 4 | 198 |
| 3 | 22 | 84 | 20 | 63 | 195 |
| 4 | 30 | 143 | 27 | 114 | 198 |
| 6 | 25 | 102 | 23 | 77 | 180 |
| 8 | 3 | 5 | 2 | 3 | 186 |
| 9 | 23 | 93 | 21 | 71 | 171 |
| 10 | 31 | 118 | 28 | 89 | 162 |
| 11 | 27 | 89 | 25 | 63 | 162 |
| 15 | 8 | 22 | 7 | 15 | 183 |
| 16 | 10 | 19 | 9 | 10 | 195 |
| 21 | 16 | 105 | 14 | 90 | 165 |
| 22 | 70 | 319 | 63 | 252 | 150 |
| 23 | 15 | 88 | 13 | 74 | 171 |
| 24 | 1 | 2 | 1 | 1 | 154 |
| 28 | 36 | 88 | 32 | 54 | 135 |
| Sum | 321 | 1285 | 289 | 980 | |

*3.2. Analysis of Variants 1–3 with Adaptation Scenarios 1–5*

This chapter analyzes the results of soil water content, total runoff, actual evapotranspiration and sediment yield for variants 1–3 along with adaptation scenarios 1–5 for the entire Bystra catchment.

From the analysis of the table for the climate change scenario RCP 4.5 (Supplementary Table S1), for individual climate change projections RCP 4.5.1, RCP 4.52 and RCP 4.5.3, it can be seen that the climate change presented in variant 1 will reduce the soil water content in the years 2041–2050. A similar situation will occur for variants 2 and 3 under adaptation scenarios V2.1 and V3.1. The greatest reduction in soil water content occurred for adaptation scenario V3.5, where the average annual soil water content decreased by up to 16% (RCP 4.5.1). The changes in soil water content may be affected by the change in crops in the study area, where the soil water content changes for a given season (V2.2–V2.5 and V3.2–V3.5) in comparison to the adaptation scenarios V2.1, V3.1 and variant 1. The reduction in soil water content, especially during the MAM and JJA seasons, can be offset by irrigation. For V2.3, V2.4, V2.5 and V3.3, the introduction of irrigation helps to offset the effects of climate change presented in variant 1 by reducing the loss of water compared to the SWAT 2010–2017 model. Similar relationships were obtained by comparing climate projections RCP 8.5.1, RCP 8.5.2 and RCP 8.5.3 (climate change scenario RCP 8.5)

(Supplementary Table S2), the difference being that for the climate change scenario RCP 8.5, the changes in soil water content in 2041–2050 is predicted to be small for variant 1 and adaptation scenarios V2.1 and V3.1 compared to the SWAT model for 2010–2017. The largest reduction in soil water content occurred for the adaptation scenario V3.5, where the average annual soil water content decreased by up to 7% (RCP 8.5.1). Changes in soil water content were also influenced by the change in crops in the study area, where the soil water content changes for a given season (V2.2–V2.5 and V3.2–V3.5) compared to adaptation scenarios V2.1, V3.1 and variant 1. Reductions in soil water content, especially during the MAM and JJA seasons, can be offset by irrigation. For V2.4 and V2.5 (RCP 8.5.1) and V2.3, V2.4, V2.5 and V3.3, the introduction of irrigation helped to offset the climate change impacts presented in variant 1. Irrigation can help to increase soil water content compared to 2010–2017 (SWAT model). The total runoff for most variants (1–3) was lower for RCP 4.5.1, RCP 4.5.2 and RCP 4.5.3 (years 2041–2050) projections compared to the 2010–2017 SWAT model (Supplementary Table S3). The exceptions were adaptation scenarios V2.4 and V2.5 for RCP 4.5.3, where the total runoff for all seasons was higher by up to 19%. The total runoff was lowest for adaptation scenario V3.5 (annual total by up to 51%) for all climate projections. In contrast, the total runoff for most variants (1–3) was higher for RCP 8.5.1, RCP 8.5.2 and RCP 8.5.3 (years 2041–2050) projections compared to the 2010–2017 SWAT model (Supplementary Table S4). The exceptions were adaptation scenarios V2.1, V2.2, V3.1, V3.2 and V3.5 and variant 1 for RCP 8.5.1, where the total runoff annual total was lower. Annual total runoff was highest for adaptation scenarios V2.4 and V2.5 (over 50%) for RCP 8.5.3. The lowest sediment yield of up to 89% for all seasons (for 2041–2050), compared to the SWAT 2010–2017 model, was for adaptation scenarios V2.5 and V3.5 for all projections (RCP 4.5.1, RCP 4.5.1 and RCP 4.5.3) (Supplementary Table S5). The lowest sediment yield occurred in the DJF season for most adaptation scenarios (RCP 4.5.1, RCP 4.5.2 and RCP 4.53) and MAM for all adaptation scenarios (RCP 4.5.1). In contrast, the highest sediment yield values occurred during the JJA season for most adaptation scenarios (RCP 4.5.1 and RCP 4.5.2). The highest annual sediment yield was as high as 35%, occurring for adaptation scenarios V2.3 and V3.3 (RCP 4.5.1, RCP 4.5.2 and RCP 4.5.3). In addition, for climate change scenario RCP 8.5 (RCP 8.5.1, RCP 8.5.2 and RCP 8.5.3), the lowest sediment yield of up to 89% for all seasons (for 2041–2050), compared to the SWAT 2010–2017 model, was for adaptation scenarios V2.5 and V3.5 (Supplementary Table S6). The lowest sediment yield occurred during the DJF season for most adaptation scenarios (RCP 4.5.2 and RCP 4.53). For most adaptation scenarios (RCP 8.5.1, RCP 8.5.2 and RCP 8.5.3), the seasonal sediment yield values were higher compared to the SWAT 2010–2017 model. In contrast, the highest sediment yield values occurred during the JJA season for RCP 8.5.1, RCP 8.5.2 and RCP 8.5.3, and the SON season for RCP 8.5.3. Annual sediment yield totals were higher for most adaptation scenarios. Actual evapotranspiration for all adaptation scenarios (RCP 4.5.1, RCP 4.5.2 and RCP 4.5.3) were higher compared to the SWAT 2010–2017 model (Supplementary Table S7). The exception was the MAM season for RCP 4.5.3, where actual evapotranspiration values were lower. For adaptation scenarios V2.3, V2.4, V2.5, V3.3, V3.4 and V3.5, associated with irrigation, actual evapotranspiration was higher compared to adaptation scenarios without irrigation (V2.1, V2.2, V.3.1 and V3.2). Similar results for actual evapotranspiration occurred for most adaptation scenarios (RCP 8.5.1, RCP 8.5.2 and RCP 8.5.3), where values were higher compared to the SWAT 2010–2017 model (Supplementary Table S8). The exception was the MAM season for RCP 8.5.2, where actual evapotranspiration values were lower. For adaptation scenarios V2.3, V2.4, V2.5, V3.3, V3.4 and V3.5, associated with irrigation, actual evapotranspiration was higher compared to adaptation scenarios without irrigation (V2.1, V2.2, V.3.1 and V3.2).

The paper also analyzed water withdrawal from reservoirs (Supplementary Table S9). The condition of reservoirs without irrigation (V3.1) was compared with the condition of reservoirs under adaptation scenarios V3.3, V3.4 and V3.5 for climate change scenarios RCP 4.5 and RCP 8.5.

Adaptation scenario V3.3 shows an increase in water withdrawals during the MAM and JJA seasons. The largest withdrawal occurs during the JJA season. Water withdrawal begins from April through to September. In the following months, reservoir resources are restored. Adaptation scenario V3.4 also shows an increase in water withdrawal. The largest withdrawal occurs during the JJA season.

In adaptation scenario V3.5, the situation is similar. The lowest water level depends on the adopted climate forecast.

A comparison of the adaptation scenarios V3.3, V3.4 and V3.5 shows that the highest water consumption is predicted to be for V3.5.

*3.3. Comparison*

This chapter compares the averages of the three RCM combinations (RACMO22E, HIRHAM5 and RCA4) under the RCP scenarios (RCP 4.5 and RCP 8.5) for the four seasons (DJF, MAM, JJA and SON).

Table 4 presents data on seasonal average soil water content for RCP 4.5 and RCP 8.5.

For RCP 4.5 (Table 4), it is apparent that for all variants, the average soil water content was lower between 2041 and 2050 compared to the SWAT 2010–2017 model. The lowest average values occurred in adaptation scenario V3.5 (13.5% decrease). The smallest decreases in soil water content occurred in the irrigation adaptation scenarios (V2.3, V2.4, V2.5 and V.3.3).

For most RCP 8.5 variants (Table 4), changes in soil water content between 2041 and 2050 were small compared to the SWAT 2010–2017 model. The only exception was for the V3.5 adaptation scenario, where it was about 7%. For adaptation scenarios V2.2, V2.3, V2.4, V2.5, V3.2, V3.3 and V3.4 for the MAM season, soil water content was higher by up to 2.5%. In contrast, for adaptation scenarios V2.4 and V2.5, the soil water content was 2% higher for the JJA season.

For the RCP 8.5 scenario, the soil water content was higher for all seasons compared to the RCP 4.5 scenario (Figure 3). For the DJF season, the soil water content was the highest. On the other hand, the lowest soil water content was found in the JJA season. For the V3.5 adaptation scenario, soil water content values for all seasons deviated significantly compared to the other adaptation scenarios.

The soil water content for RCP 4.5 in the years 2041–2050 varied from 290 mm (V2.2 and V3.2, JJA) to 336 mm (V3.3, DJF) (Table 4 and Figure 3). For the SWAT 2010–2017 model, the soil water content varied from 309 mm (JJA) to 344 mm (DJF) (Table 4 and Figure 3). The exception was V3.5, where the soil water content varied from 269 mm (JJA) to 313 mm (DJF). The change in crops in V2.2 and V3.2 scenarios for RCP 4.5 appears to increase the soil water content (up to 321–322 mm, MAM) when compared to the BaU (V1) scenario, where the soil water content was 310 mm (2041–2050). The situation was different for the JJA season, with the soil water content falling from 295 mm (V1) to 290 mm (V2.2 and V3.2). However, for most scenarios with irrigation (V2.3–V2.5, V3.3 and V3.4 for RCP 4.5) the soil water content was 314–324 mm (MAM). Whilst for the BaU (V1) scenario, the soil water content was 310 mm (2041–2050). The JJA season was similar, with a soil water content of 296 mm (except for V3.4) to 305 mm. For V1, in the JJA season, the soil water content was 295 mm.

**Table 4.** Average soil water content in the Bystra catchment for DJF (December, January, February, etc.), MAM, JJA and SON in 2041–2050 under various scenarios (SWAT model, V1, V2.1-V2.5 and V3.1-V3.5) for climate scenarios RCP 4.5 and RCP 8.5. Numbers in red (decrease) or blue (increase) refer to a percentage change in water content in the soil (bold numbers) (author's own study).

| Time Interval | | 2041–2050 | | | | | | | | | | |
| --- | --- | --- | --- | --- | --- | --- | --- | --- | --- | --- | --- | --- |
| | | | Variant 2: Small Retention, More Ponds | | | | | | Variant 3: Small Retention, More Reservoirs | | | |
| Type of Scenario | Model 2010-2017 | Variant 1 Only Climate Change (V1) | Cereals (V2.1) | Vegetables (V2.2) | Irrigated Vegetables (V2.3) | Irrigated Vegetables + Cereals (V2.4) | Irrigated Orchard + Cereals (V2.5) | Cereals (V3.1) | Vegetables (V3.2) | Irrigated Vegetables (V3.3) | Irrigated Vegetables + Cereals (v3.4) | Irrigated Orchard + Cereals (V3.5) |
| **Season** **Climate Scenario** | | Seasonal Average of Soil Water Content (mm) **RCP 4.5** | | | | | | | | | | |
| DJF | 344 | 335 −2.6% | 335 −2.7% | 335 −2.7% | 335 −2.5% | 335 −2.6% | 335 −2.6% | 335 −2.6% | 335 −2.6% | 336 −2.5% | 335 −2.8% | 313 −9.1% |
| MAM | 322 | 310 −3.5% | 310 −3.5% | 321 −0.1% | 323 +0.5% | 316 −1.7% | 317 −1.5% | 310 −3.5% | 322 0.0% | 324 +0.6% | 314 −2.3% | 301 −6.5% |
| JJA | 309 | 295 −4.4% | 295 −4.5% | 290 −6.3% | 298 −3.6% | 305 −1.2% | 305 −1.3% | 295 −4.4% | 290 −6.2% | 296 −4.3% | 294 −4.7% | 269 −12.9% |
| SON | 328 | 319 −2.7% | 318 −2.9% | 316 −3.7% | 318 −2.9% | 320 −2.2% | 321 −2.0% | 319 −2.7% | 316 −3.5% | 318 −2.9% | 317 −3.3% | 284 −13.5% |
| **Average Annual** | 326 | 315 −3.3% | 315 −3.4% | 315 −3.2% | 319 −2.1% | 319 −2.0% | 319 −1.9% | 315 −3.3% | 316 −3.1% | 318 −2.3% | 315 −3.2% | 292 −10.5% |
| **Climate Scenario** | | **RCP 8.5** | | | | | | | | | | |
| DJF | 344 | 340 −1.1% | 340 −1.1% | 340 −1.3% | 340 −1.4% | 340 −1.1% | 340 −1.2% | 340 −1.1% | 341 −1.0% | 341 −1.0% | 340 −1.1% | 330 −4.3% |
| MAM | 322 | 320 −0.5% | 320 −0.5% | 327 +1.5% | 326 +1.4% | 324 +0.7% | 324 +0.9% | 320 −0.5% | 328 +2.0% | 330 +2.5% | 323 +0.4% | 317 −1.4% |
| JJA | 309 | 308 −0.1% | 308 −0.3% | 298 −3.4% | 305 −1.4% | 315 +2.0% | 315 +2.0% | 308 −0.2% | 299 −3.3% | 305 −1.3% | 307 −0.7% | 290 −6.0% |
| SON | 328 | 328 +0.1% | 328 0.0% | 327 −0.2% | 328 +0.3% | 329 +0.3% | 329 +0.4% | 328 +0.1% | 326 −0.6% | 327 −0.2% | 327 −0.2% | 305 −6.8% |
| **Average Annual** | 326 | 324 −0.4% | 324 −0.5% | 323 −0.8% | 325 −0.3% | 327 +0.4% | 327 +0.5% | 324 −0.4% | 323 −0.7% | 326 0.0% | 324 −0.4% | 311 −4.6% |

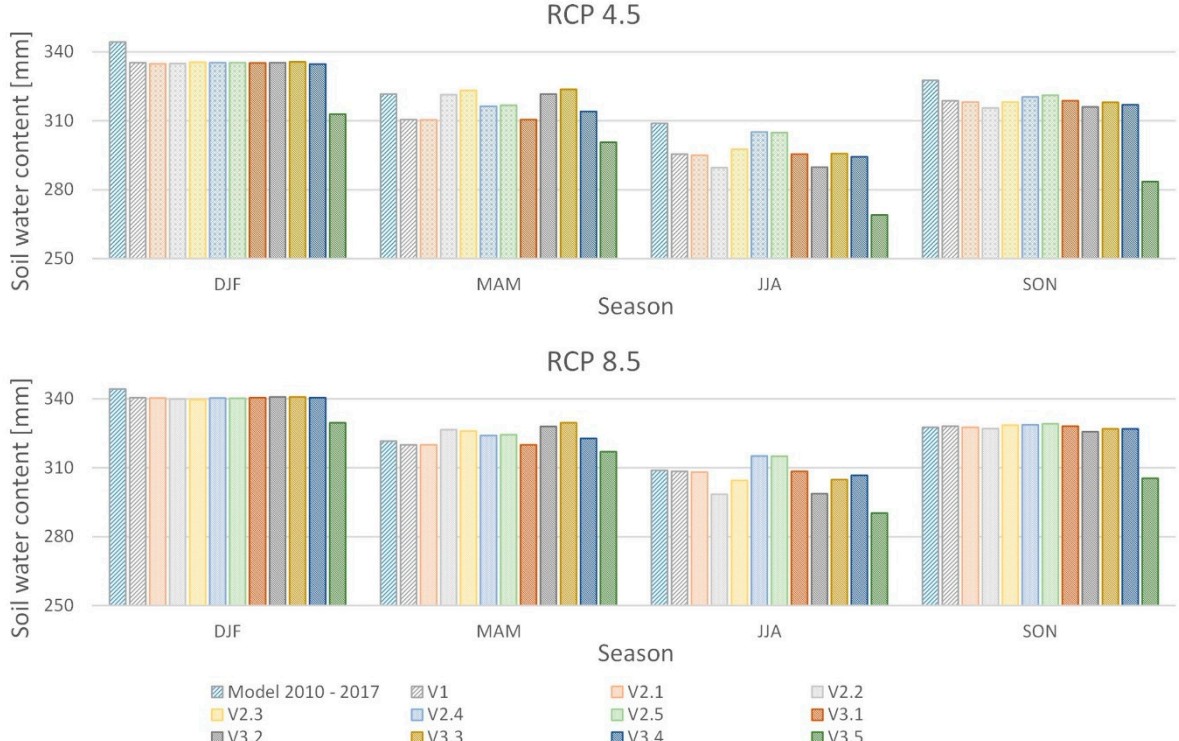

**Figure 3.** Comparison of the average distribution of soil water content by season for the SWAT 2010–2017 simulation period with variants 1–3 for the RCP 4.5 and RCP 8.5 climate change scenarios, for the years 2041–2050 in the Bystra catchment (author's own study).

The soil water content for RCP 8.5 in the years 2041–2050 ranged from 298 mm (V2.2 and V3.2, JJA) to 341 mm (V3.2 and V3.3, DJF). The exception was V3.5, where the soil water content ranged from 290 mm (JJA) to 330 mm (DJF). However, for most scenarios with irrigation (V2.3–V2.5, V3.3 and V3.4 for RCP 8.5) the soil water content ranged from 323 mm to 330 mm (MAM). Whilst for the BaU (V1) scenario, the soil water content was 320 mm (2041–2050). In the JJA season, the soil water content was 315 mm (V2.4 and V2.5). For V1, in the JJA season the soil water content was 308 mm. The remaining scenarios with irrigation will display smaller soil water content in the JJA season when compared to V1. As was the case for RCP 4.5, a change in crop can contribute to the increase in the soil water content.

Total runoff for the RCP 4.5 scenario (2041–2050) decreased for most variants (Table 5) compared to the SWAT 2010–2017 model. The largest decrease in total runoff occurred for adaptation scenario V3.5. Small decreases in total runoff occurred for adaptation scenarios V2.3, V2.4, V2.5 and V3.3. Adaptation scenarios V2.4 and V2.5 experienced an increase in total runoff (up to 6%) during the JJA and SON seasons.

All RCP 8.5 scenario variants (2041–2050) presented an increase in total runoff compared to the SWAT 2010–2017 model (Table 5). The highest total runoff occurred in adaptation scenarios V2.4 and V2.5 (up to 43%). The lowest total runoff occurred in adaptation scenario V3.5.

The total runoff graphs for RCP 4.5 and RCP 8.5 (Figure 4) show a higher total runoff for all seasons in the RCP 8.5 scenario. For the V3.5 adaptation scenario, the total runoff values for all seasons deviated significantly compared to the other adaptation scenarios.

**Table 5.** Average total runoff in the Bystra catchment for DJF (December, January, February, etc.), MAM, JJA and SON in 2041–2050 in various scenarios (SWAT model, V1, V2.1-V2.5 and V3.1-V3.5) for climate scenarios RCP 4.5 and RCP 8.5. Numbers in red (decrease) or blue (increase) refer to a percentage change in water content in the soil (bold numbers) (author's own study).

| Time Interval | | 2041–2050 | | | | | | | | | | |
|---|---|---|---|---|---|---|---|---|---|---|---|---|
| | | | Variant 2: Small Retention, More Ponds | | | | | | Variant 3: Small Retention, More Reservoirs | | | |
| Type of Scenario | Model 2010-2017 | Variant 1 Only Climate Change (V1) | Cereals (V2.1) | Vegetables (V2.2) | Irrigated Vegetables (V2.3) | Irrigated Vegetables + Cereals (V2.4) | Irrigated Orchard + Cereals (V2.5) | Cereals (V3.1) | Vegetables (V3.2) | Irrigated Vegetables (V3.3) | Irrigated Vegetables + Cereals (v3.4) | Irrigated Orchard + Cereals (V3.5) |
| Season | | | | | Seasonal Total of Total Runoff (mm) | | | | | | | |
| Climate Scenario | | | | | RCP 4.5 | | | | | | | |
| DJF | 55 | 42 −24% | 41 −25% | 40 −27% | 46 −16% | 51 −7% | 52 −4% | 42 −24% | 40 −27% | 45 −18% | 43 −21% | 35 −35% |
| MAM | 54 | 42 −23% | 41 −23% | 41 −23% | 46 −14% | 50 −7% | 52 −4% | 42 −23% | 42 −23% | 46 −15% | 43 −20% | 36 −33% |
| JJA | 46 | 35 −24% | 35 −25% | 35 −24% | 42 −9% | 47 +3% | 49 +6% | 35 −24% | 35 −24% | 41 −11% | 38 −18% | 31 −34% |
| SON | 48 | 38 −22% | 37 −23% | 35 −28% | 43 −11% | 50 +4% | 51 +6% | 38 −22% | 35 −27% | 41 −14% | 39 −18% | 32 −34% |
| Annual Total | 202 | 156 −23% | 154 −24% | 150 −26% | 177 −13% | 198 −2% | 203 0% | 156 −23% | 152 −25% | 173 −15% | 163 −19% | 134 −34% |
| Climate Scenario | | | | | RCP 8.5 | | | | | | | |
| DJF | 55 | 59 +8% | 59 +7% | 55 +0% | 61 +13% | 67 +24% | 68 +26% | 59 +8% | 56 +3% | 62 +14% | 61 +11% | 53 −3% |
| MAM | 54 | 59 +9% | 58 +8% | 58 +7% | 62 +15% | 66 +23% | 68 +26% | 59 +9% | 58 +8% | 63 +17% | 60 +13% | 54 0% |
| JJA | 46 | 54 +17% | 53 +16% | 52 +13% | 61 +32% | 65 +40% | 66 +43% | 54 +17% | 53 +14% | 60 +30% | 57 +23% | 50 +9% |
| SON | 48 | 56 +17% | 56 +16% | 54 +12% | 59 +22% | 67 +40% | 68 +42% | 56 +17% | 53 +10% | 60 +25% | 59 +22% | 51 +7% |
| Annual Total | 202 | 228 +13% | 226 +12% | 218 +8% | 243 +20% | 266 +31% | 271 +34% | 228 +13% | 220 +9% | 245 +21% | 236 +17% | 208 +3% |

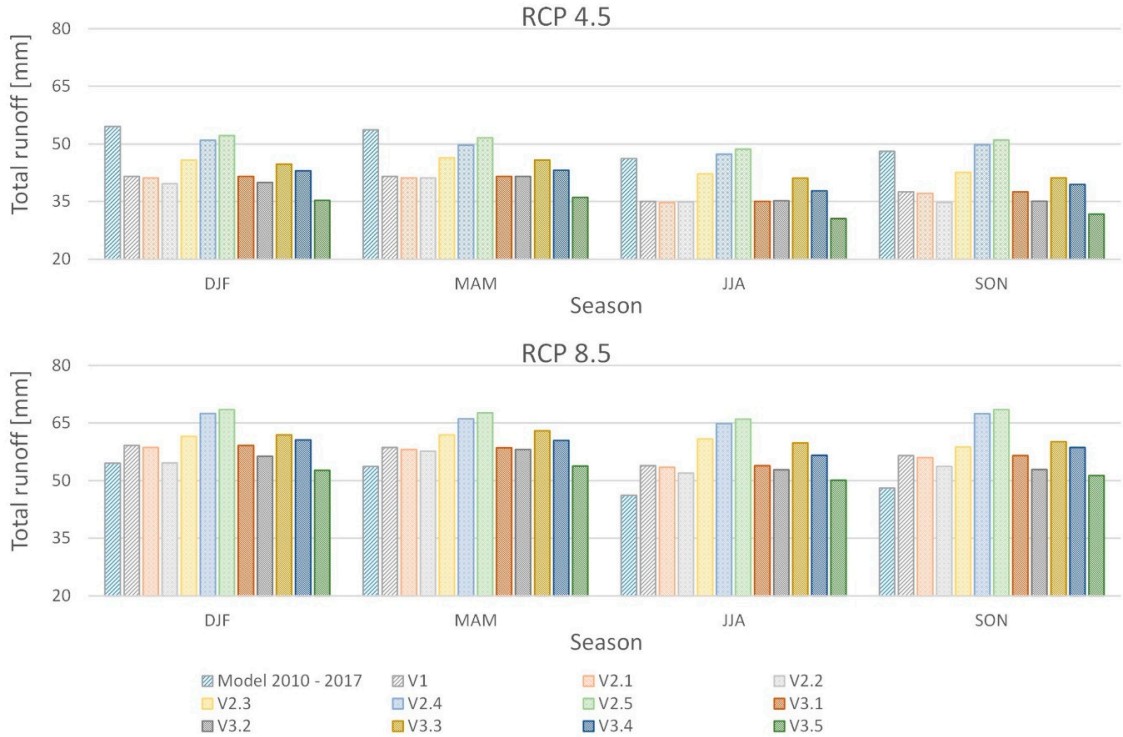

**Figure 4.** Comparison of the average distribution of total runoff by season for the SWAT 2010–2017 simulation period with variants 1–3 for the RCP 4.5 and RCP 8.5 climate change scenarios, for the years 2041–2050 in the Bystra catchment (author's own study).

The total runoff was lower (between 31 mm and 46 mm) (V1, V2.1–V2.3 and V3.1–V3.5) for the 2041–2050 period when compared to the SWAT 2010–2017 model (between 46 mm and 55 mm) for all seasons in the RCP 4.5 scenario (Table 5 and Figure 4). On the other hand, for V2.4 and V2.5, the total runoff ranged from 47 mm to 52 mm. An increase in the total runoff occurred in the JJA and SON seasons (V2.4 and V2.5). The total runoff for RCP 8.5 ranged from 50 mm to as much as 68 mm (V2.5).

Furthermore, it was observed in most scenarios with irrigation (V2.3, V2.4, V2.5, V3.3 and V3.4) that the annual total runoff increases, e.g., values of 163–203 mm (RCP 4.5) and 236–271 mm (RCP 8.5) were higher when compared to V1 (BaU) (156 mm (RCP 4.5) and 228 mm (RCP 8.5) in 2041–2050. However, for the scenarios in which land use was changed into vegetable farming (V2.2 and V3.2), the total runoff was lower (150–152 mm (RCP 4.5) and 218–220 mm (RCP 8.5)).

For RCP scenario 4.5 (2041–2050), the largest decreases in sediment yield (up to 86% compared to the SWAT 2010–2017 model), occurred for adaptation scenarios V2.5 and V3.5 (Table 6). However, increases in sediment yields compared to the SWAT 2010–2017 model occurred for adaptation scenarios V2.3, V2.4, V3.2, V3.3 and V3.4 (up to 116%). For all adaptation scenarios for the DJF season, there was a decrease in sediment yield. In contrast, for JJA there was an increase in sediment yield for most adaptation scenarios.

There was an increase in sediment yield (up to 226%) for most adaptation scenarios (in most seasons) for RCP 8.5 (2041–2050) compared to the SWAT 2010–2017 model (Table 6). The exception was a decrease in sediment yield (up to 80%) for adaptation scenarios V2.5 and V3.5.

**Table 6.** Average sediment yield in the Bystra catchment for DJF (December, January, February, etc.), MAM, JJA and SON in 2041–2050 under various scenarios (SWAT model, V1, V2.1-V2.5 and V3.1-V3.5) for climate scenarios RCP 4.5 and RCP 8.5. Numbers in red (decreases) or blue (increases) refer to a percentage change in the water content in soil (bold numbers) (author's own study).

| Time Interval | | 2041–2050 | | | | | | | | | | | |
|---|---|---|---|---|---|---|---|---|---|---|---|---|---|
| | | | **Variant 2: Small Retention, More Ponds** | | | | | | **Variant 3: Small Retention, More Reservoirs** | | | | |
| Type of Scenario | Model 2010-2017 | Variant 1 Only Climate Change (V1) | Cereals (V2.1) | Vegetables (V2.2) | Irrigated Vegetables (V2.3) | Irrigated Vegetables + Cereals (V2.4) | Irrigated Orchard + Cereals (V2.5) | Cereals (V3.1) | Vegetables (V3.2) | Irrigated Vegetables (V3.3) | Irrigated Vegetables + Cereals (v3.4) | Irrigated Orchard + Cereals (V3.5) |
| **Season** **Climate Scenario** | | Seasonal Total of Sediment Yield (t/ha) | | | | | RCP 4.5 | | | | | |
| DJF | **0.21** | 0.13 −38% | 0.11 −49% | 0.16 −25% | 0.16 −24% | 0.13 −35% | 0.03 −86% | 0.13 −38% | 0.19 −9% | 0.19 −9% | 0.16 −23% | 0.04 −83% |
| MAM | **0.18** | 0.12 −36% | 0.10 −48% | 0.21 +16% | 0.22 +22% | 0.19 +1% | 0.03 −82% | 0.12 −36% | 0.26 +42% | 0.27 +46% | 0.21 +14% | 0.04 −78% |
| JJA | **0.11** | 0.13 +22% | 0.10 −2% | 0.16 +53% | 0.22 +109% | 0.21 +97% | 0.03 −70% | 0.13 +22% | 0.19 +84% | 0.23 +116% | 0.17 +62% | 0.04 −64% |
| SON | **0.15** | 0.18 +19% | 0.15 −2% | 0.11 −29% | 0.15 0% | 0.15 0% | 0.04 −77% | 0.18 +19% | 0.13 −13% | 0.15 +1% | 0.13 −13% | 0.04 −71% |
| **Annual Total** | **0.65** | 0.56 −14% | 0.45 −30% | 0.64 −2% | 0.75 +16% | 0.68 +5% | 0.13 −80% | 0.56 −14% | 0.77 +20% | 0.84 +29% | 0.67 +4% | 0.16 −76% |
| **Climate Scenario** | | | | | | | RCP 8.5 | | | | | |
| DJF | **0.21** | 0.17 −18% | 0.14 −33% | 0.16 −23% | 0.22 +4% | 0.18 −16% | 0.04 −80% | 0.17 −18% | 0.27 +32% | 0.28 +33% | 0.21 +2% | 0.05 −76% |
| MAM | **0.18** | 0.16 −15% | 0.13 −30% | 0.29 +57% | 0.29 +57% | 0.24 +32% | 0.04 −77% | 0.16 −15% | 0.34 +85% | 0.36 +93% | 0.28 +51% | 0.05 −72% |
| JJA | **0.11** | 0.19 +81% | 0.15 +47% | 0.22 +112% | 0.34 +226% | 0.26 +147% | 0.05 −55% | 0.19 +81% | 0.27 +160% | 0.31 +197% | 0.24 +126% | 0.06 −43% |
| SON | **0.15** | 0.32 +112% | 0.26 +73% | 0.29 +93% | 0.23 +54% | 0.23 +54% | 0.07 −54% | 0.32 +112% | 0.30 +101% | 0.32 +112% | 0.25 +64% | 0.09 −44% |
| **Annual Total** | **0.65** | 0.84 +29% | 0.68 +6% | 0.96 +49% | 1.08 +67% | 0.91 +40% | 0.20 −69% | 0.84 +29% | 1.19 +84% | 1.26 +95% | 0.97 +50% | 0.25 −62% |

The sediment yield graphs for RCP 4.5 and RCP 8.5 (Figure 5) show higher sediment yield values for all seasons for the RCP 8.5 scenario. For the V2.5 and V3.5 adaptation scenarios, the sediment yield values for all seasons deviated significantly compared to the other adaptation scenarios. Sediment yield for RCP 4.5 in the years 2041–2050 varied from 0.10 t/ha to 0.27 t/ha for most scenarios. For the SWAT 2010–2017 model, the sediment yield varied from 0.11 t/ha to 0.21 t/ha (Table 6 and Figure 5). V2.5 and V3.5 are different, as their sediment yield ranges from 0.03 t/ha to 0.04 t/ha. However, the sediment yield for RCP 8.5 in the years 2041–2050 varied from 0.13 t/ha to 0.36 t/ha for most scenarios except for V2.5 and V3.5, where it ranges from 0.04 t/ha to 0.09 t/ha.

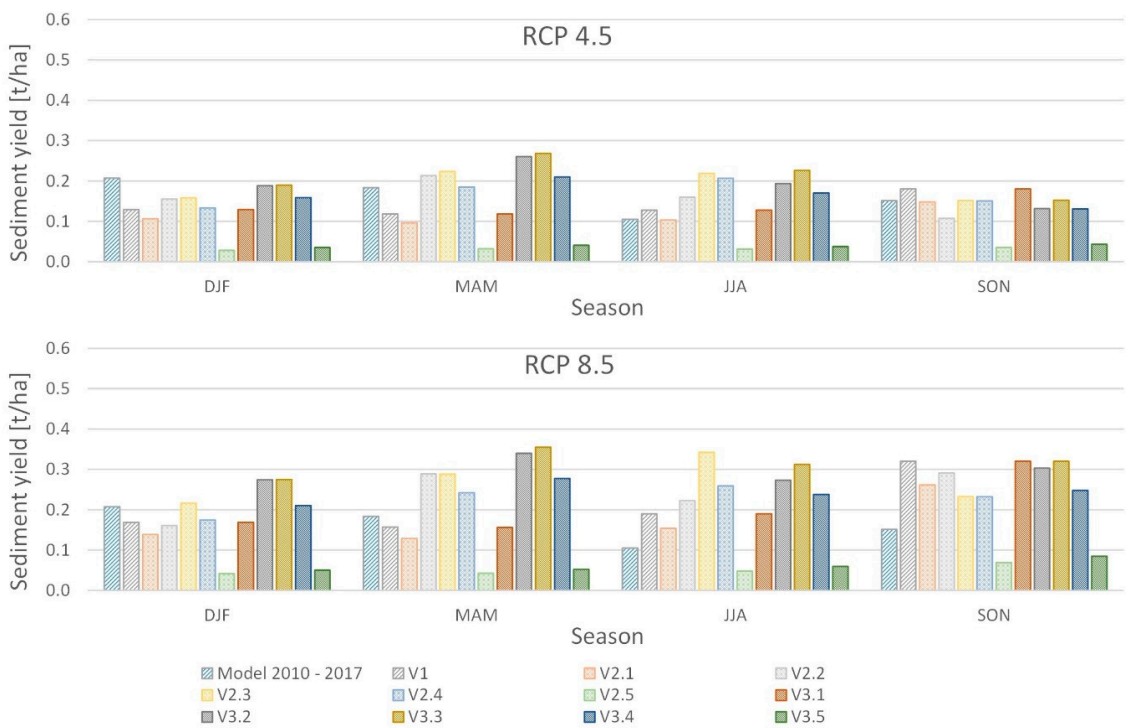

**Figure 5.** Comparison of the average distribution of sediment yield by season for the SWAT 2010–2017 simulation period with variants 1–3 for the RCP 4.5 and RCP 8.5 climate change scenarios, for the years 2041–2050 in the Bystra catchment (author's own study).

Moreover, most scenarios with irrigation (V2.3, V2.4, V3.3 and V3.4) showed an increase in the annual value of the sediment yield, of 0.67–0.84 t/ha (RCP 4.5) and 0.91–1.26 t/ha (RCP 8.5) compared to V1 (BaU) (0.56 t/ha (RCP 4.5) and 0.84 t/ha (RCP 8.5)) in the years 2041–2050. An increase was also observed in the scenarios which changed the land use into vegetable farming (V2.2 and V3.2) (0.64–0.77 t/ha (RCP 4.5) and 0.96–1.19 t/ha (RCP 8.5)).

In the RCP 4.5 and RCP 8.5 scenarios (Figure 6), the actual evapotranspiration values of each variant were similar (Table 7). Actual evapotranspiration for all adaptation scenarios for 2041–2050 was higher compared to the SWAT 2010–2017 model. The exception was the MAM season (RCP 4.5 and RCP 8.5), where actual evapotranspiration values were lower for V2.2, V2.3, V3.3 and V3.4 (up to 15%), and season JJA (RCP 4.5 and RCP 8.5), where actual evapotranspiration values were lower for V1, V2.1 and V3.1 (up to 5%). For adaptation scenarios V2.3, V2.4, V2.5, V3.3, V3.4 and V3.5, which are associated with irrigation, actual evapotranspiration was higher compared to adaptation scenarios without irrigation (V2.1, V2.2, V3.1 and V3.2).

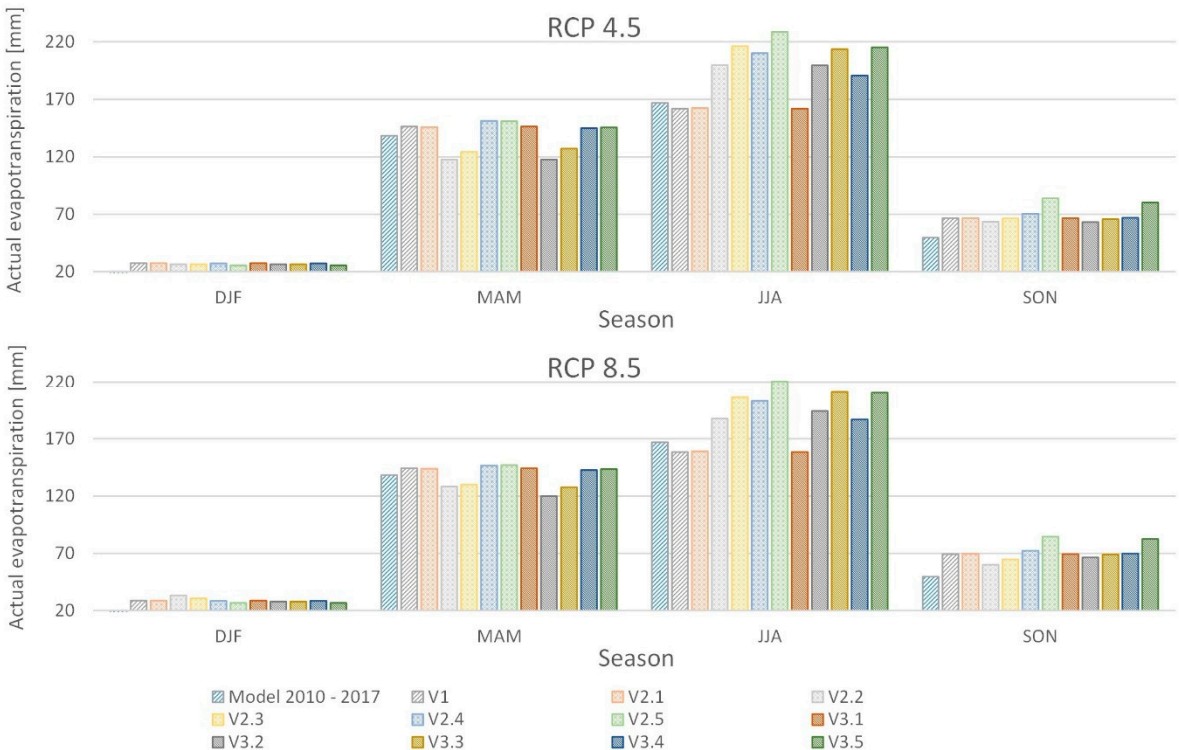

**Figure 6.** Comparison of the average distribution of actual evapotranspiration by season for the SWAT 2010–2017 simulation period with variants 1–3 for the RCP 4.5 and RCP 8.5 climate change scenarios, for the years 2041–2050 in the Bystra catchment (author's own study).

For RCP 4.5 and RCP 8.5, the highest values of actual evapotranspiration occurred in the JJA season, i.e., from 158 mm to 229 mm (Table 7 and Figure 6), for the years 2041–2050. For most scenarios, these values were higher when compared to the SWAT 2010–2017 model (167 mm). However, a slight increase up to 151 mm was observed for the MAM season. For the SWAT 2010–2017 model, the evapotranspiration was 138 mm. Lower values were noted for the V2.2, V2.3, V3.2 and V3.3 scenarios (from 118 mm to 130 mm). In addition, in scenarios with irrigation (V2.3, V2.4, V2.5, V3.3, V3.4 and V3.5) for RCP 4.5 and RCP 8.5, it was observed that the annual value of evapotranspiration increased (428–489 mm) when compared to V1 (BaU) (401–402 mm) in the years 2041–2050. An increase was also observed in the scenarios where the land use was changed into vegetable farming (V2.2 and V3.2), from 407 mm to 409 mm.

**Table 7.** Average actual evapotranspiration in the Bystra catchment for DJF (December, January, February, etc.), MAM, JJA and SON in 2041–2050 under various scenarios (SWAT model, V1, V2.1-V2.5 and V3.1–V3.5) for climate scenarios RCP 4.5 and RCP 8.5. Numbers in red (decreases) or blue (increasea) refer to a percentage change in the water content in soil (bold numbers) (author's own study).

| Time Interval | | 2041–2050 | | | | | | | | | | | |
| --- | --- | --- | --- | --- | --- | --- | --- | --- | --- | --- | --- | --- | --- |
| | | | | Variant 2: Small Retention, More Ponds | | | | | | Variant 3: Small Retention, More Reservoirs | | | |
| Type of Scenario | Model 2010-2017 | Variant 1 Only Climate Change (V1) | Cereals (V2.1) | Vegetables (V2.2) | Irrigated Vegetables (V2.3) | Irrigated Vegetables + Cereals (V2.4) | Irrigated Orchard + Cereals (V2.5) | Cereals (V3.1) | Vegetables (V3.2) | Irrigated Vegetables (V3.3) | Irrigated Vegetables + Cereals (v3.4) | Irrigated Orchard + Cereals (V3.5) |
| **Season Climate Scenario** | | Seasonal Total of Actual Evapotranspiration (mm) RCP 4.5 | | | | | | | | | | |
| DJF | 17 | 27 +60% | 27 +59% | 26 +53% | 26 +53% | 27 +58% | 26 +49% | 27 +60% | 26 +53% | 26 +53% | 27 +58% | 26 +49% |
| MAM | 138 | 146 +6% | 146 +6% | 118 −15% | 124 −10% | 151 +9% | 151 +9% | 146 +6% | 118 −15% | 127 −8% | 145 +5% | 146 +5% |
| JJA | 167 | 162 −3% | 162 −3% | 200 +20% | 216 +29% | 210 +26% | 229 +37% | 162 −3% | 200 +20% | 214 +28% | 191 +14% | 215 +29% |
| SON | 50 | 67 +35% | 67 +35% | 63 +28% | 66 +34% | 70 +42% | 84 +69% | 67 +35% | 63 +28% | 66 +33% | 67 +35% | 80 +62% |
| **Annual Total** | 372 | 402 +8% | 403 +8% | 407 +9% | 433 +16% | 459 +23% | 489 +31% | 402 +8% | 407 +9% | 433 +16% | 430 +16% | 467 +26% |
| **Climate Scenario** | | RCP 8.5 | | | | | | | | | | |
| DJF | 17 | 29 +68% | 29 +68% | 33 +93% | 31 +78% | 29 +66% | 27 +55% | 29 +68% | 28 +62% | 28 +62% | 29 +67% | 27 +56% |
| MAM | 138 | 144 +4% | 144 +4% | 128 −7% | 130 −6% | 147 +6% | 147 +6% | 144 +4% | 120 −13% | 127 −8% | 143 +3% | 143 +4% |
| JJA | 167 | 158 −5% | 159 −5% | 188 +12% | 206 +23% | 203 +22% | 221 +32% | 158 −5% | 194 +16% | 211 +26% | 187 +12% | 210 +26% |
| SON | 50 | 69 +40% | 69 +40% | 60 +21% | 65 +30% | 72 +46% | 85 +70% | 69 +40% | 66 +34% | 69 +39% | 70 +41% | 82 +66% |
| **Annual Total** | 372 | 401 +8% | 401 +8% | 409 +10% | 431 +16% | 450 +21% | 479 +29% | 401 +8% | 409 +10% | 435 +17% | 428 +15% | 463 +24% |

## 4. Discussion

The results related to the water content in the soil were compared with the values of water capacity and wilting point obtained from a previous water balance study in the research basin [2]. The data on soils in the catchment area of the Bystra River were prepared on the basis of the above-mentioned study. For a 1.5 m soil profile, the results of the above-mentioned studies are consistent with this publication.

The aim of the paper was to analyze the results of soil water content (1.5 m profile), actual evapotranspiration, total runoff and sediment yield for the three variants and to evaluate them against the current state of knowledge related to studies involving similar adaptation studies. Variants 1–3 were derived from simulations of a calibrated and validated SWAT model for three regional climate models derived from the EC-EARTH global climate model for 2041–2050 [12]. Variant 1 (V1) includes only climate change. Variant 2 includes pond design for each farm in the catchment. In contrast, Variant 3 includes the design of water reservoirs. Variants 2 and 3 each include five adaptation scenarios.

The first adaptation scenario (V2.1 and V3.1) involves only increasing the number of ponds on the farm or increasing the number of reservoirs for non-irrigated arable land crops, i.e., WWHT (winter cereals), BARL (spring cereals), CANP (rapeseed) and CRDY (other crops). The second adaptation scenario (V2.2 and V3.2) involves growing vegetables without irrigation (instead of cereals). The third adaptation scenario (V2.3 and V3.3) involves growing vegetables with irrigation (instead of cereals). The fourth adaptation scenario (V2.4 and V3.4) involves partial cultivation of vegetables and cereals. The fifth adaptation scenario (V2.5 and V3.5) involves partial cultivation of orchards and cereals.

Soil water content for different climate change projections (RCP 4.5.1, RCP 4.5.2, RCP 4.5.3, RCP 8.5.1, RCP 8.5.2 and RCP 8.5.3) were compared in a paper on projected soil water content in 2050 using adaptation scenarios involving changes in land use and agricultural practices [28]. Between 2041 and 2050, average annual soil water content was 3.3% lower for the RCP 4.5 scenario compared to the RCP 8.5 scenario (0.4%) [28]. Under most seasons, soil water content was lower (up to 4.4%) compared to the 2010–2017 model.

For the years 2041–2050, the soil water content was higher for the RCP 8.5 scenario than for the RCP 4.5 scenario. The RCP 8.5 scenario has slight differences in soil water content compared to the 2010–2017 simulation.

Visual estimates of soil water content (green water) were compared between this study and the studies on the European SWAT model [8] and the SWAT model for the Vistula and the Oder [70]. Due to the lack of possibility to compare the numerical values and to check the depth of the soil profile by visual assessment, it can be concluded that the water storage in the soil (in the area of the Bystra catchment) for the study area of Europe was within the range of 130–160 mm, while for the basin of the Vistula and the Oder it was in the range of 300–350 mm. The average annual soil water content for the Bystra catchment in 2010–2017 was 326 mm. This value is comparable with the data obtained from the study on the Vistula and Oder catchments.

A study on the model and projection of soil moisture anomalies and yield losses in Poland was also analyzed [71]. This study used a set of nine error-adjusted EURO-CORDEX projections for two future horizons: 2021–2050 and 2071–2100 under two scenarios: RCP4.5 and 8.5. The article showed that in the years 2021–2050 and 2071–2100, the forecasts indicate an intensification of water shortages in the soil for the main crops of spring cereals, potatoes and maize. Moreover, for the historical period, it was shown that the deficit in soil water content leading to yield losses occurred more frequently than excess water content in soil. The study noted that the water content of the soil was dependent on the weather, but also on the physical properties and the quality of the soil.

This study also found lower water levels in the soil in the coming decades.

A comprehensive review of pond, wetland and reservoir tools compiled in the article on the rock catchment, which is a tributary of the Mississippi in the United States [72], showed that the type of retention had little effect on the simulation of the flow of the stream, but had a significant effect on the modeling of sediment transport.

In an article on the evaluation of the effectiveness of automatic irrigation of the Texas High Plains, attention was drawn to the continuous decrease in the appropriate water level from the Ogallala reservoir, which is experiencing long-term pumping of irrigation water [73]. It was found that the irrigation method based on the occurrence of water stress related to water deficit in the soil does not adequately reflect field practices due to the continuation of irrigation after the crop is mature (over-irrigation). This study indicated the need to refine or create a more sensitive and intuitive automatic irrigation algorithm in the SWAT program to better match the irrigation parameters to the real needs of growing plants in the irrigated region.

Another article looked at the influence of climate on the irrigation process of the Alentejo catchment area in southern Portugal in the Mediterranean region [74]. This article described adaptation measures to combat climate changes in the future such as taking water from the Monte Novo and Vigo reservoirs for the purpose of irrigating crops, increasing the efficiency of water use in irrigation systems, and changing some crops.

These publications describing both the Texas High Plains and Alentejo in southern Portugal also indicated the problems in replenishing the resources of water reservoirs, resulting from the location of these reservoirs in places where the rebuilding of water resources takes much longer than in Poland (higher temperature and less precipitation). In the studied area of the Bystra catchment area, in the coming decades, water resources will not have problems related to the deficit of water for irrigation.

Changes in soil water content for vegetables and cereals, which were presented in the adaptation scenarios, may be due to the difference in water uptake from the soil resulting from the specifications of these plants. For both cereals and vegetables, hydration should be optimized to fully meet the water needs of vegetables during their growth. Again, there is a need to improve the irrigation algorithm in SWAT, as in the study from Texas High Plains [73]. This study indicated the need to refine or create a more sensitive and intuitive automatic irrigation algorithm in the SWAT program to better match the irrigation parameters to the real needs of growing plants in the irrigated region.

Qualitative environmental data and managerial practices clearly affect the SWAT simulation process and the related processes in the catchment, including plant growth. Irrigation management is critical to the quantification of changes in the hydrological constituents and crop responses in various automatic irrigation scenarios.

Kellya et al. [75] analyzed the choice of irrigation planning strategy and its impact on water use and profits. They showed that effective irrigation planning is essential to improve agricultural water efficiency and manage the negative impacts of agricultural water abstraction on other water users and on the environment. This planning should also consider helping farmers to assess and develop an appropriate irrigation planning strategy for their specific production conditions. Where these strategies are not adapted to local agronomic and biophysical conditions, the potential increase in profitability and efficiency in water use may be too great.

Various irrigation systems are currently being tested in Poland, including the ENORA-SIS system, i.e., Environmental Optimization of Irrigation Management with the Combined Use and Integration of High Precision Satellite Data, Advanced Modelling, Process Control and Business Innovation [76]. This system is designed to support decisions for the sustainable irrigation of crops. The role of the extensive ENORASIS system is to support the farmer in making decisions about the amount of irrigation and when the given irrigation should take place. The system was tested in 2014 on a potato crop. Optimal conditions for the plant to develop were made possible by adjusting the soil moisture level and the needs of the crop, resulting in maximized yield and greater disease resistance. The improved results of ENORASIS were compared to the results obtained with standard irrigation. This optimized irrigation also showed how water and energy is wasted with non-optimized irrigation [77].

Variant 2 involved the creation of multiple ponds on agricultural land in the Bystra catchment. There are many studies describing this issue [72], but none of them examine the

effect of designed ponds on soil water storage. Most studies examine the effects of ponds on erosion and sediment transport, such as the study of the Claise catchment in France [78]. As shown in the above study, erosion risk as well as sediment yield also increased in scenarios without ponds. In the Bystra catchment, increasing the number of ponds had little effect on soil water content and actual evapotranspiration (on comparison of adaptation scenarios V1 and V2.1). However, increasing the number of ponds contributed to a decrease in sediment yield losses and total runoff (Table 6, Supplementary Tables S15 and S16) (on comparison of adaptation scenarios V1 and V2.1).

Changes in soil water content and sediment yield are strongly affected by changes in agricultural cultivation or land use. The change from cereal cultivation (V1, V2.1 and V3.1) to vegetable cultivation (V2.2, V2.3, V2.4, V3.2, V3.3 and V3.4) or orchard (V2.5 and V3.5) changed the soil water content. However, such a change did not significantly affect the total runoff (except for in irrigated crops). A study on the effects of climate change and vegetation [79] showed that changing the vegetation in an area can have a significant impact on soil moisture. A similar study looked at the change in land use from forest to rice fields, where it was shown to have an effect on soil moisture [80].

In the emissive RCP 4.5 scenario, the precipitation in the years 2041–2050 was lower by 4–8% for all seasons (DJF, MAM and JJA) except SON (precipitation higher by 7%) when compared to the SWAT 2010–2017 model. On the other hand, in the emissive RCP 8.5 scenario, the precipitation in the years 2041–2050 was higher by 1–25% for all seasons [12] (Supplementary Table S10). Similar results have been observed in most tropical basins in Africa [81] and Latin America [82–84]. Additionally, precipitation was found to have a strong impact on the hydrological response of a tropical basin in Colombia. These results converge with other studies carried out under similar conditions [85–88]. According to these mentioned studies, the future water supply in tropical basins will be much smaller than now. These findings agree with studies in the tropics [89,90], which claim that the assumption that the future water availability will be equal to that of today's is unreliable in the changing world. Reality puts at least two challenges before bodies managing water resources: firstly, implementing a management strategy based on scenarios; secondly, suggesting adaptation strategies to alleviate potential climate change effects. In this sense, a major advantage of hydrological models is studying the consequences of various conditions and alternatives for water management.

The lack of influence of the increase in the number of ponds as well as water reservoirs on some components of the water balance in the catchment area may be due to certain limitations of the SWAT model. However, the possibility of using water for irrigation from retention reservoirs presented in this article could possibly also work for ponds.

Agriculture is largely related to, and can significantly affect, climatic conditions. An increase in the occurrence of unfavorable weather conditions in consecutive decades may have a serious impact on the quality and volume of farmed crops. According to previous studies, in the future, we may face declining amounts of soil water during the growing season, which is shown in the climate change scenarios [12]. Climate change may also entail other threats, such as droughts, heavy precipitation, erosion [91], floods, strong winds and landslides [17]. To carry out beneficial changes in the landscape over a larger area, appropriate tools must be used. In Poland, such tools are land consolidations, which are supported by appropriate legal regulations with a scheme of conduct developed over many years. Land consolidations in an extended scope allow the implementation of beneficial changes in the landscape (afforestation, retention reservoirs and irrigation) in terms of water retention [91–93]. There are several agricultural scientific centers in Poland working on the issue of managing rural areas, i.e., IUNG-PIB in Puławy and University of Agriculture in Kraków, among others. Many years of research have resulted in a scheme for comprehensive management of rural areas (KUOW in Polish) [93,94]. KUOW primarily deals with land consolidation, water management, village renewal, landscape design and wildlife conservation. Importantly, KUOW comprises water management including water and anti-erosion drainage; designating areas for anti-flood investments, including small

retention to control flood water as well as keeping soil water on fields during droughts; anti-erosion measures; and building a network of small mills on watercourses that could contribute to renewable energy. Moreover, some of the tasks of landscape design and wildlife conservation are protecting soil by anti-erosion shaping of the stolon and creating buffer zones to protect waters [94]. The Comprehensive Management of Rural Areas recommends the most effective way of introducing a spectrum of changes in the rural landscape. Based on the results of this study, the team is planning a simulation of a comprehensive scenario using KUOW as one option towards a better holistic management of water in rural areas.

Introducing small retention reservoirs (ponds and storage reservoirs) into the catchment landscape does not translate directly to a retention or an increase in water supply in a catchment. The soil water content slightly decreased from 314.9 mm for V1 to 314.6 mm for V2.1 (RCP 4.5, 2041–2050), and from 324.2 mm for V1 to 324.0 mm for V2.1 (RCP 8.5, 2041–2050) (Supplementary Tables S13 and S17).

For V3.1, the decrease in content is predicted to be c. 0.01%. The smaller soil water content in the variant featuring ponds could be caused by the uptake (drawdown) of water by the ponds in the surrounding area.

The fact that the designed ponds and storage reservoirs seem to have little impact on some components of the water balance of the Bystra catchment can be attributed to the imperfections in the SWAT model. Our study points at the need to refine the model or design a more sensitive and user-friendly algorithm in the SWAT program so that the changes in parameters of ponds and reservoirs are better reflected in the water balance of the catchment.

Notwithstanding the above, the differences in the changes in water balance in the catchment can be found by way of deduction based on the changes in the flow, the sediment outflow, the total runoff and the sediment yield. The outflow (m$^3$/s) in MAM, JJA and SON seasons was lower for the selected catchments (Supplementary Table S11) for the V2.1 variant when compared to V1 variant (RCP 4.5 and RCP 8.5) in the years 2041–2050. However, it was higher for the DJF season. For the subcatchment area 31 (Bystra river mouth into the Vistula), in RCP 4.5, in the JJA season the outflow decreased from 1.44 (V1) to 1.42, and in the SON season it decreased from 1.45 to 1.40. In the DJF season, it increased from 1.55 to 1.62. However, in RCP 8.5, in the JJA season the outflow decreased from 2.17 (V1) to 2.15, and in the SON season it decreased from 2.14 to 2.11. In DJF, it increased from 2.21 to 2.25. For V3.1, the annual outflow will increase from 1.51 to 1.53 (RCP 4.5) and from 2.19 to 2.21 (RCP 8.5) (Supplementary Table S11). Generally, the higher findings for V3.1 in individual subcatchment areas could be a result of a larger water discharge from retention reservoirs.

The sediment outflow was lower in 2041–2050 for most subcatchments in the RCP 4.5 and RCP 8.5 scenarios for V2.1 when compared to V1 (Supplementary Table S12). For subcatchment 31 (Bystra river mouth into the Vistula), the annual sum decreased from 2241 (metric tons) to 2022 (metric tons) (RCP 4.5) and from 3260 to 2865 (RCP 8.5). For V3.1, the decrease in the sediment outflow was visible mainly in subcatchment 31, where the annual sum decreased from 2241 (metric tons) to 1237 (metric tons) (RCP 4.5) and from 3260 to 2084 (RCP 8.5).

The total runoff decrease for V2.1 is presented in the results of the subcatchment (Supplementary Table S15). For each selected subcatchment, the total runoff decreases by up to several per cent. However, considering the whole catchment area, the decrease in total runoff was about 1% for RCP 4.5 and RCP 8.5 (Supplementary Table S17). The annual sum decreased from 155.6 mm (V1) to 154.1 mm (V2.1) for RCP 4.5 and from 228.1 mm (V1) to 226.1 (V2.1) for RCP 8.5. No considerable changes were noted for V3.1.

Just as in the case of the total runoff, the sediment yield also showed a decrease for V2.1 (Supplementary Table S16). For all subcatchments, a decrease in sediment yield was noted. In terms of the whole catchment area, the sediment yield decrease was c. 18% for RCP 4.5 and RCP 8.5 (Supplementary Table S17). The annual sum decreased from 0.56 t/ha

(V1) to 0.45 t/ha (V2.1) for RCP 4.5 and from 0.84 t/ha (V1) to 0.68 t/ha (V2.1) for RCP 8.5. No considerable changes were noted for V3.1.

Due to the decreases in sediment yield, sediment outflow and total runoff (Supplementary Table S12), designing ponds can also be useful when it comes to trapping nitrogen, phosphorus compounds and other pollution before they enter the water flow. Secondly, the possibility of surface water erosion will be smaller. Similar preventive features can be achieved by implementing non-plough cultivation and designing filtering zones [28].

The pond design was based on dispersing ponds (non-irrigated arable lands CRDY acc. to CLC) of dimensions 50 m × 100 m (5000 sq m) and 3 m depth in individual subcatchments. The ponds were designed in compliance with the Polish construction laws that allow building a pond larger than 1000 sq m, but no more than 5000 sq m, and up to 3 m deep. It was assumed that one pond would be built for every two farms. The average area of an individual farm with more than 1 ha of agricultural land in 2015 in Lubelskie Voivodeship was 8 ha. The combined area of ponds in the catchment is 752 ha, which constitutes 2.7% of the catchment area, including the existing reservoirs and watercourses. Before the project it was 0.3% [12].

Large retention reservoirs, however, were designed on the watercourses at the existing mouths of the catchments. As building such huge reservoirs is extremely complex, the choice of their site and design was simplified.

The designing of ponds (in some cases designating areas for bigger ponds) during land consolidation, as well as their construction and arrangement during post-consolidation works, has—according to this study—scientific grounds for implementation in the coming decades, with the aim of increasing water resources in a landscape and consequently making crops more resilient to droughts. Compared to retention reservoirs, they can be a cheaper alternative. The means of financing the construction of a pond can be indicated during the process of preparation for land consolidation. Moreover, ponds can be used as water sources for irrigation.

Similar to non-ploughed land cultivation [28], variants featuring the construction of ponds and retention reservoirs can also contribute to reducing the negative impact of precipitation deficits in the Bystra catchment. It is particularly significant in the RCP 4.5 emission scenario (Supplementary Table S10), where lower precipitation was projected in 2041–2050 ([12], the first article concerning water balance in the Bystra). Furthermore, reducing the negative impact of a precipitation deficit will be particularly important in the MAM and JJA seasons, as these are the plant growing seasons. Some variants (V2.3–V2.5 and V3.3–V3.5) use irrigation, which also reduces the negative effect of a precipitation deficit. Another adaptation is the implementation of crops that are less vulnerable to droughts while being economically justified. This requires further research.

In terms of uncertainties related to data quality influencing the simulation outputs, we have used the best available dataset: 1:25000 digital soil map, 5 m DEM derived from LIDAR and good quality land use based on orthophoto digitization and CLC. A study carried out by Bouslihim et al. [95] presented the operation/running of the SWAT model using two different soil databases in order to assess the impact of the quality of soil data on the hydrological behavior and to analyze the water balance in the catchment. The first database contained a low-resolution soil map of three types of soil collected from the Harmonized World Soil Database produced by FAO. The other database contained a high-resolution soil map of eleven types of soil, compiled by land surveys and laboratory analyses. The results demonstrated that the quality and the resolution of a soil map affects the number of HRU units, while modification of some parameters, such as the depth of soil, affects the components of a hydrological cycle, including soil water content and actual evapotranspiration. The annual sum of actual evapotranspiration and the average annual soil water content in the low resolution soil model were, respectively, 141.46 mm and 13.69 mm, while in the high-resolution model they were 163.04 mm and 17.36 mm.

### 5. Conclusions

In the RCP 4.5 scenario, the precipitation in the years 2041–2050 is predicted to be lower by 4–8% for all seasons (DJF, MAM and JJA), except for SON (precipitation is predicted to be higher by 7%) compared to the SWAT 2010–2017 model. On the other hand, in the RCP 8.5 scenario, the precipitation in the years 2041–2050 is predicted to be higher by 1–25% for all seasons [12] (Supplementary Table S10).

The decrease in the soil water content for RCP 4.5 in 2041–2050, compared to the SWAT 2010–2017 model, is predicted to fluctuate between 0.1% and 6.3% (V2.2 and V3.2) (Table 4 and Supplementary Table S10). An exception is V3.5, where the drop in soil water content fluctuates between 6.5% (MAM) and 13.5% (SON). A c. 1% increase in the soil water content is predicted in V2.3 and V3.3 scenarios in the MAM season. The change in crop (e.g., into vegetables) may contribute to an increase in the soil water content.

The V2.2 and V3.2 scenarios show an increase in soil water content to 321–322 mm (MAM) when compared to the BaU (V1) scenario, which has a soil water content of 310 mm (2041–2050). The situation is different for the JJA season, where the soil water content drops from 295 mm (V1) to 290 mm (V2.2 and V2.3). For RCP 8.5, the drop in soil water content is predicted to fluctuate between 0.1% and 3.4% (V2.2 and V3.2) (Table 4 and Supplementary Table S10). An exception is V3.5, where the decrease in soil water content fluctuates between 1.4% (MAM) and 6.8% (SON). The increase in soil water content is predicted to fluctuate between 0.1% and 2.5%. The increase in soil water content occurs for most variants in the MAM season. As was the case for RCP 4.5, the change in crop may contribute to an increase in the soil water content.

Variants 2 and 3 with irrigation scenarios from deep aquifers or engineered reservoirs (V2.3 and V3.3) contribute to the increased soil water content (especially during the JJA season) compared to adaptation scenarios V2.2 and V3.2. For RCP 4.5, in the JJA season, the soil water content increases from 290 mm (V2.2 and V3.2) to 296–298 mm (V2.3 and V3.3). However, for RCP 8.5, in the JJA season, the soil water content increases from 298–299 mm (V2.2 and V3.2) to 305 mm (V2.3 and V3.3) (Table 4).

Changes in soil water content and sediment yield are strongly influenced by changes in agricultural cultivation or land use. The change from cereal cultivation (V1, V2.1 and V3.1) to vegetable cultivation (V2.2, V2.3, V2.4, V3.2, V3.3 and V3.4) or orchards (V2.5 and V3.5) altered the soil water content. For RCP 4.5, in the MAM season (2041–2050), in most scenarios (V2.2–V2.5 and V3.2–V3.4), an increase in the soil water content was observed (from 314 mm to 324 mm) when compared to the scenarios without a change in crop (V1, V2.1 and V3.1), where the soil water content was 310 mm. A slightly smaller growth is seen in the JJA season. For V1, V2.1 and V3.1 the soil water content is 295 mm, while for V2.3–V2.5 and V3.3 it is from 296 mm to 305 mm. For RCP 8.5, in the MAM season (2041–2050), in most scenarios (V2.2–V2.5 and V3.2–V3.4), an increase in the soil water content was also observed (from 323 mm to 330 mm) when compared to the scenarios without a change in crop (V1, V2.1 and V3.1), where the soil water content was 320 mm. On the other hand, in the JJA season, a growth was observed only for V2.4 and V2.5 (315 mm) when compared to V1, V2.1 and V3.1 (308 mm) (Table 4).

For the MAM and JJA seasons, soil water content is reduced for all variants compared to the DJF season, which may be due to uptake of soil water by plants. For RCP 4.5 (2041–2050), in the DJF season, the soil water content is 335–336 mm for most scenarios (V1, V2.1–V2.5 and V3.1–V3.4), in the MAM season, the soil water content is 310–324 mm, while in the JJA season, it is 290–305 mm. In the SON season, the soil water content is 316–321 mm. Similarly, for RCP 8.5, in the DJF season, the soil water content is 340–341 mm for most scenarios (V1, V2.1–V2.5 and V3.1–V3.4), in the MAM season, the soil water content is 320–330 mm, in the JJA season it is 298–315, while in the SON season it is 326–329 mm (Table 4).

The highest values of actual evapotranspiration were found in the JJA season (Table 7 and Supplementary Table S10). Its changes are predicted to fluctuate between −3% (V1, V2.1 and V3.1) and 37% (V2.5) for the years 2041–2050 (MAM and JJA) when compared to

the SWAT 2010–2017 model. However, a drop in actual evapotranspiration was observed for the MAM season, ranging from 8% (V3.3) to 15% (V2.2 and V3.2). For RCP 8.5, the highest values of actual evapotranspiration are predicted to occur in the JJA season, as was the case for the RCP 4.5 scenario. Its changes are predicted to fluctuate between −5% (V1, V2.1 and V3.1) and 32% (V2.5) (MAM and JJA). A drop in actual evapotranspiration is predicted to occur for the MAM season, reaching from 6% (V2.3) to 13% (V3.2) (Table 7).

What is more, the scenarios featuring irrigation (V2.3, V2.4, V2.5, V3.3, V3.4 and V3.5) for RCP 4.5 and RCP 8.5 show an increase in the annual value of evapotranspiration (428–489 mm) when compared to V1 (401–402 mm) in the years 2041–2050 (Table 7).

For adaptation scenarios V2.3, V2.4, V2.5, V3.2, V3.4 and V3.5, which are associated with irrigation, actual evapotranspiration is predicted to be higher compared to adaptation scenarios without irrigation (V2.1, V2.2, V.3.1 and V3.2).

The total runoff for RCP 4.5 is predicted to be smaller by 22% to 35% (V1, V2.1, V2.2, V3.1, V3.2 and V3.5) for the 2041–2050 decade as compared to the SWAT 2010–2017 model for all seasons (Table 5 and Supplementary Table S10). However, for V2.3, V2.4, V2.5 and V3.3, the total runoff is predicted to decrease from −18% to +6%. An increase in the total runoff is predicted to occur in JJA and SON seasons (V2.4 and V2.5). The total runoff for RCP 8.5 is predicted to increase in most scenarios, by up to 43% (V2.5) in the JJA season.

Moreover, most scenarios featuring irrigation (V2.3, V2.4, V2.5, V3.3 and V3.4) for RCP 4.5 and RCP 8.5 show an increase in total runoff when compared to V1 in the years 2041–2050 (Table 5 and Supplementary Table S10).

Sediment yield for RCP 4.5 is predicted to be highest in the JJA season for V2.4 (97%), V2.3 (109%) and V3.3 (116%) in the years 2041–2050 when compared to the model period. However, a fall is predicted to occur in the DJF season amounting to between 9 and 49% (V2.1) (Table 6 and Supplementary Table S10), with the exception of V2.5 and V3.5 scenarios, where there is a fall in sediment yield between 64% and 86% for all seasons. For RCP 8.5, the sediment yield is predicted to be highest in the JJA season for V3.3 (197%) and V2.3 (226%) in 2041–2050 when compared to the model period 2010–2017. However, the value is predicted to drop in the DJF season, amounting to 33% (V2.1) for most variants, with the exception of V2.5 and V3.5 scenarios, where for all seasons the sediment yield is predicted to drop between 43% and 80%.

Additionally, the V2.5 and V3.5 scenarios for RCP 4.5 and RCP 8.5 show a considerable drop in annual total sediment yield (0.13–0.25 t/ha) when compared to V1 (0.56–0.84 t/ha) in the years 2041–2050 (Table 6).

Orchard adaptation scenarios (V2.5 and V3.5) reduce the sediment yield. The annual sum of sediment yield decreases from 0.65 t/ha for the SWAT 2010–2017 model to 0.13–0.25 t/ha for V2.5 and V3.5 in the RCP 4.5 and RCP 8.5 scenarios. Additionally, the V3.5 adaptation scenario significantly reduces the soil water content in all seasons. The average annual soil water content falls from 326 mm for the SWAT 2010–2017 model to 292 mm (RCP 4.5) and 311 mm (RCP 8.5) for V3.5.

In the Bystra catchment, increasing the number of ponds had little effect on soil water content and actual evapotranspiration (on comparison of adaptation scenarios V1 and V2.1). However, increasing the number of ponds contributed to a decrease in sediment yield losses and total runoff (on comparison of adaptation scenarios V1 and V2.1). The volume of ponds (V2.1) in this paper is more than twice ($2258 \text{ m}^3 \cdot 10^4$) the volume of water reservoirs ($1065 \text{ m}^3 \cdot 10^4$ and $1094 \text{ m}^3 \cdot 10^4$) in the Bystra catchment (Supplementary Table S9). Ponds are dispersed all over the area of the catchment while water reservoirs are set on the river in its different reaches. Supplying ponds with rainfall or spring water cannot be relied on, but water reservoirs are mainly supplied by a flowing river. However, an analysis should be performed on whether supplying a designed reservoir from the water course in a given section will be enough to fill it. It is worth mentioning that building ponds should not be very expensive as they can be built on inexpensive land belonging to farmers. On the other hand, building large reservoirs is a complex and expensive engineering enterprise that requires buying land for investment. Land consolidation could be a solution. In addition,

ponds can be planned during land consolidation works, making the whole investment even more economical. A disadvantage of large reservoirs is the costs and technical problems connected with building a suitable irrigation infrastructure in the furthermost parts of the catchment. On the plus side, however, large water reservoirs are structures that are more resistant to the changing hydrological and weather conditions in the catchment compared to ponds. They can also serve a recreational purpose and ameliorate the landscape and the environment.

A failure to observe a clear impact of the designed ponds and retention reservoirs (comparing V1, V2.1 and V3.1 in 2050) on some components of the water balance of the Bystra catchment may be attributed to the imperfection of the SWAT model.

Our study indicates the need to refine the model or design a more sensitive and user-friendly algorithm in the SWAT program so that the changes in parameters of ponds and reservoirs are better reflected in the water balance of the catchment.

The designing of ponds (in some cases designating areas for bigger ponds) during land consolidation, as well as their construction and arrangement during post-consolidation works, has—according to this study—scientific grounds for implementation in the coming decades, with the aim of increasing water resources in a landscape and consequently making crops more resilient to droughts. Compared to retention reservoirs they can be a cheaper alternative. The means of financing the construction of a pond can be indicated during the process of preparation for land consolidation. Moreover, ponds can be used as water sources for irrigation. They can be used to reduce the sediment outflow and prevent erosion (decreasing the total runoff). Increasing the area of ponds to 2.7% (from 0.3% BaU) lowers the total runoff by c. 1% (RCP 4.5 and RCP 8.5) and decreases the sediment yield by c. 18% (RCP 4.5 and RCP 8.5). The sediment outflow shows also a tendency to decrease in most watercourses in the catchment.

It seems a future small retention policy should therefore be a hybrid using the possibility to build ponds (even larger than 5000 sq m) wherever possible. However, in order to maintain the hydrological stability of the region, resistant to the changing climate conditions, large reservoirs should also be built in suitable locations.

With lower precipitation levels in the years 2041–2050, relative to 2010–2017, as presented in the emissive scenario RCP 4.5, the soil water content decreases by up to 14% for most variants. Total runoff for most variants will also be lower by 4–35%. The percentage change in sediment yield will fluctuate between −86% and 116%. On the other hand, the actual evapotranspiration in most variants is predicted to be higher.

With higher precipitation levels in the years 2041–2050, relative to 2010–2017, as presented in the emissive scenario RCP 8.5, the soil water content changes slightly from −7% to +3%. Total runoff for most variants is also predicted to be higher by as much as 43%. Sediment yield for most scenarios may increase by 226%. The actual evapotranspiration for most variants is also predicted to be higher. The biggest increase in soil water content is found in most irrigation variants for RCP 4.5 (annual average 316–319 mm) (V2.3–V2.5, V3.2 and V3.3) and RCP 8.5 (annual average 326–327 mm) (V2.3–V2.5 and V3.3) when compared to V1 (BaU) (315 mm for RCP 4.5 and 324 mm for RCP 8.5) for the years 2041–2050. The smallest increase in soil water content, however, is found in the V3.5 variant: the annual average was 292 mm for RCP 4.5 and the annual average was 311 mm for RCP 8.5. For the future changes in climate, irrigation comprising of water reservoirs (ponds or storage reservoirs) is a solution worth considering in Bystra river catchment.

**Supplementary Materials:** The following supporting information can be downloaded at: https://www.mdpi.com/article/10.3390/agronomy13020404/s1, Table S1. Comparison of average soil water content by season between model SWAT 2010–2017 and V1, V2.1-V2.5 and V3.1-V3.5 for 2041–2050 in the Bystra catchment for climate projection RCP 4.5.1, RCP 4.5.2, RCP 4.5.3. Bold numbers indicate soil water content, and shaded numbers indicate percentage change (red indicates % decrease in content and blue indicates % increase in content). Dark red and dark blue shading indicates large changes, while light red and light blue shading indicates small changes (author's own study). Table S2. Comparison of average soil water content by season between model SWAT 2010–2017 and V1, V2.1-

V2.5 and V3.1-V3.5 for 2041–2050 in the Bystra catchment for climate projection RCP 8.5.1, RCP 8.5.2, RCP 8.5.3. Bold numbers indicate soil water content, and shaded numbers indicate percentage change (red indicates % decrease in content and blue indicates % increase in content). Dark red and dark blue shading indicates large changes, while light red and light blue shading indicates small changes (author's own study). Table S3. Comparison of seasonal total runoff between model SWAT 2010–2017 and V1, V2.1-V2.5 and V3.1-V3.5 for 2041–2050 in the Bystra catchment for climate projection RCP 4.5.1, RCP 4.5.2, RCP 4.5.3. Bold numbers indicate total runoff, and shaded numbers indicate percentage change (red indicates % decrease in content and blue indicates % increase in content). Dark red and dark blue shading indicates large changes, while light red and light blue shading indicates small changes (author's own study). Table S4. Comparison of seasonal total runoff between model SWAT 2010–2017 and V1, V2.1-V2.5 and V3.1-V3.5 for 2041–2050 in the Bystra catchment for climate projection RCP 8.5.1, RCP 8.5.2, RCP 8.5.3. Bold numbers indicate total runoff, and shaded numbers indicate percentage change (red indicates % decrease in content and blue indicates % increase in content). Dark red and dark blue shading indicates large changes, while light red and light blue shading indicates small changes (author's own study). Table S5. Comparison of seasonal sediment yield between model SWAT 2010–2017 and V1, V2.1-V2.5 and V3.1-V3.5 for 2041–2050 in the Bystra catchment for climate projection RCP 4.5.1, RCP 4.5.2, RCP 4.5.3. Bold numbers indicate sediment yield, and shaded numbers indicate percentage change (red indicates % decrease in content and blue indicates % increase in content). Dark red and dark blue shading indicates large changes, while light red and light blue shading indicates small changes (author's own study). Table S6. Comparison of seasonal sediment yield between model SWAT 2010–2017 and V1, V2.1-V2.5 and V3.1-V3.5 for 2041–2050 in the Bystra catchment for climate projection RCP 8.5.1, RCP 8.5.2, RCP 8.5.3. Bold numbers indicate sediment yield, and shaded numbers indicate percentage change (red indicates % decrease in content and blue indicates % increase in content). Dark red and dark blue shading indicates large changes, while light red and light blue shading indicates small changes (author's own study). Table S7. Comparison of seasonal actual evapotranspiration between model SWAT 2010–2017 and V1, V2.1-V2.5 and V3.1-V3.5 for 2041–2050 in the Bystra catchment for climate projection RCP 4.5.1, RCP 4.5.2, RCP 4.5.3. Bold numbers indicate actual evapotranspiration, and shaded numbers indicate percentage change (red indicates % decrease in content and blue indicates % increase in content). Dark red and dark blue shading indicates large changes, while light red and light blue shading indicates small changes (author's own study). Table S8. Comparison of seasonal actual evapotranspiration between model SWAT 2010–2017 and V1, V2.1-V2.5 and V3.1-V3.5 for 2041–2050 in the Bystra catchment for climate projection RCP 8.5.1, RCP 8.5.2, RCP 8.5.3. Bold numbers indicate actual evapotranspiration, and shaded numbers indicate percentage change (red indicates % decrease in content and blue indicates % increase in content). Dark red and dark blue shading indicates large changes, while light red and light blue shading indicates small changes (author's own study). Table S9. Comparison of water withdrawals from irrigation reservoirs for adaptation scenarios V3.3, V3.4, V3.4 in relation to the total amount of water in reservoirs without irrigation (V3.1) in the Bystra catchment from 2041 to 2050 for climate scenarios RCP 4.5 and RCP 8.5. Bold numbers indicate water withdrawals from irrigation reservoirs, and shaded numbers indicate percentage change (red indicates % decrease in content) (author's own study). Table S10. Percentage summary of changes in precipitation, soil water content, sediment yield, total runoff and actual evapotranspiration under V1, V2.1-V2.5, V3.1-V3.5 in the years 2041–2050 compared to the SWAT 2010–2017 model scenario, for average values of three GCMs/RCMs combinations under two RCP climate change scenarios (RCP 4.5, RCP 8.5). The summary covers four seasons (DJF, MAM, JJA, SON) in the Bystra catchment. Shaded numbers indicate percentage changes (red indicates % decrease in content, and blue indicates % increase in content) (author's own study). Table S11. Comparison of average water flowout from the selected sub-catchments (1, 4, 9, 10, 22, 31) by season between Variant 1 (V1 - BaU) and Variants 2.1 and 3.1 for the years 2041–2050 in the Bystra catchment for climate scenarios RCP 4.5 and RCP 8.5. Bold numbers indicate outflow (m$^3$/s) and shaded numbers indicate percentage change. Shade of red indicates % decrease in content and shade of blue indicates % increase in content (author's own elaboration). Table S12. Comparison of average sediment outflow from the selected sub-catchments (1, 4, 9, 10, 22, 31) by season between Variant 1 (V1-BaU) and Variants 2.1 and 3.1 for the years 2041–2050 in the Bystra catchment for climate scenarios RCP 4.5 and RCP 8.5. Bold numbers indicate sediment outflow (metric tonnes) and shaded numbers indicate percentage change. Shade of red indicates % decrease in content and shade of blue indicates % increase in content (author's own elaboration). Table S13. Comparison of mean actual evapotransspiration in the selected sub-catchment (1, 4, 9,

10, 22, 31) by season between Variant 1 (V1-BaU) and Variants 2.1 and 3.1 for the years 2041–2050 in the Bystra catchment for climate scenarios RCP 4.5 and RCP 8.5. Bold numbers indicate actual evapotransspiration (mm) and shaded numbers indicate percentage change. Shade of red indicates % decrease in content and shade of blue indicates % increase in content (author's own elaboration). Table S14. Comparison of average soil water content in the selected sub-catchment (1, 4, 9, 10, 22, 31) by season between Variant 1 (V1-BaU) and Variants 2.1 and 3.1 for the years 2041–2050 in the Bystra catchment for climate scenarios RCP 4.5 and RCP 8.5. Bold numbers indicate soil water content (mm) and shaded numbers indicate percentage change. Shade of red indicates % decrease in content and shade of blue indicates % increase in content (author's own elaboration). Table S15. Comparison of mean total runoff in the selected sub-catchment (1, 4, 9, 10, 22, 31) by season between Variant 1 (V1-BaU) and Variants 2.1 and 3.1 for the years 2041–2050 in the Bystra River catchment for climate scenarios RCP 4.5 and RCP 8.5. Bold numbers indicate total runoff (mm) and shaded numbers indicate percentage change. Shade of red indicates % decrease in content and shade of blue indicates % increase in content (author's own elaboration). Table S16. Comparison of average sediment yield in the selected sub-catchment (1, 4, 9, 10, 22, 31) by season between Variant 1 (V1-BaU) and Variants 2.1 and 3.1 for 2041–2050 in the Bystra catchment for climate scenarios RCP 4.5 and RCP 8.5. Bold numbers indicate sediment yield (t/ha) and shaded numbers indicate percentage change. Shade of red indicates % decrease in content and shade of blue indicates % increase in content (author's own elaboration). Table S17. Summary of the average: actual evapotranspiration (mm), soil water content (mm), total runoff (mm) and sediment yield (t/ha) for the entire Bystra catchment by season between Variant 1 (V1-BaU) and Variants 2.1 and 3.1 for the years 2041–2050 for climate scenarios RCP 4.5 and RCP 8.5. Bold numbers indicate the corresponding value and shaded numbers indicate the percentage change. Shade of red indicates % decrease in content and shade of blue indicates % increase in content (author's own elaboration).

**Author Contributions:** Conceptualization, D.B. and R.W.; methodology, D.B and R.W.; software, D.B.; validation, D.B. and R.W.; formal analysis, R.W. and A.K-B.; investigation, D.B.; resources, D.B.; data curation, R.W. and A.K.-B.; writing—original draft preparation, D.B.; writing—review and editing, A.K-B..; visualization, D.B.; supervision, R.W.; project administration, R.W.; funding acquisition, R.W. All authors have read and agreed to the published version of the manuscript.

**Funding:** Research and ACP was funded by Polish Ministry of Agriculture and Rural Development, DC2.2 Shaping soil retention as an element of counteracting agricultural drought and supporting rational water management.

**Data Availability Statement:** Not applicable.

**Conflicts of Interest:** The authors declare no conflict of interest.

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
