# Peer review of "Modelling 2050 Water Retention Scenarios for Irrigated and Non-Irrigated Crops for Adaptation to Climate Change Using the SWAT Model: The Case of the Bystra Catchment, Poland"

_agronomy, doi:10.3390/agronomy13020404_

Round 1

Reviewer 1 Report

This manuscript discusses the results of a study of the impact of climate change on the hydrology and sediment transport in the Bystra catchment using SWAT and various climate change scenarios. My key concern is that the presentation of the results basically consists of a series of thirteen tables, which results in a flood of information that makes it impossible to see the forest for the trees. The presentation would be much improved if the authors were selective about the information they want to present in the main paper and discuss those parts in detail, while relegating most of the information in Tables 4 to 16 to a supplementary information section.

A second concern is that the conclusions are highly qualitative in nature, resulting in statement such as “Variants 2 and 3 with irrigation scenarios (V2.3, V3.3) from deep aquifers or engineered reservoirs contribute to the increased soil water content” which is just saying that irrigation makes the soil wetter. The paper would be very much improved if this and similar statements in the conclusions could be formulated in a quantitative way, for example by stating that an x% increase in precipitation results in y% increase of soil moisture content.

Lastly, overall the manuscript is well written, but there are some spots where the language does need to be improved. I have attached a version of the manuscript with suggestions for edits and other minor points.

Reviewer 2 Report

Materials and methods section is divided into 7 subsections instead of 5 (line 142)

I would appreciate to see the soil and land use map to understand the distribution of these two spatial information.

The source of the necessary soil parameters has not been mentioned in the text (OM, texture, depth...).

Please take into consideration the direct effect of the soil parameters on the model results and for that I propose the following article: https://doi.org/10.1016/j.jafrearsci.2019.103616

The SWAT model used in this study was calibrated over a period of 5 years (2010-2014) and validated for 3 years (2015-2017) (2.6. SWAT CUP calibration and validation results). On the other hand, the available climate data are between 2005 and 2017 (2.5. Meteorogical data). So why not the SWAT model has not been calibrated over a long period to capture the maximum climate variation?

This work is similar to other works published by the same authors, can you give explanations to that?

Reviewer 3 Report

Taking the small Bystra catchment in the east of Poland as a case study area, the paper presents the estimated changes of the soil water content, the total runoff, the sediment yield and the actual evapotranspiration of 2041-2050. The paper analyses three variants of small retention with respective adaptation scenarios for farming practices.

The paper is mainly well organized. However, the model parameters are not given clear. Some of the figures are not clear enough. Paragraphs can be organized more succinctly. The table presenting the simulation results needs further refinement. Please find below some defects which need to be further revised.

1. Lines 37-140: There are many paragraphs in the article. Please reduce and merge the same topic paragraphs.

2. Lines 162-169: Please provide the land use and soil type map of the study area.

3. Line 234: Please explain the missing meteorological data in Table 1, such as the temperature data of Rogalów station.

4. Table 4-16: It seems that these table structures could be improved, please consider carefully.

5. Lines 626-804: Please combine the research results of this paper to discuss and simplify the content.

Reviewer 4 Report

I have read with interest your paper on Modelling 2050 water retention scenarios as adaptation to climate change, using the SWAT model: The case of the Bystra catchment, Poland. The manuscript deals with very interesting problem. All stages of investigation are clearly and logically presented, thesis are proved by data, tables and illustrations. It results in interesting ‘discussion’ and clear, however, a bit too short ‘conclusions’. Therefore I have a few comments only.

l. 142-146 – introduction to subsection counter is not necessary, especially 5 subsections was introduced whereas 7 was enumerated

Chapters (sections) 3, 4, & 5 should be transformed into subsections of the ‘Results’ section for more clear layout. Please note that ‘summary’ as a main section in the middle of the manuscript and long before ‘conclusions’ looks a bit strange. Moreover, in my opinion, more suitable title for subsection ‘summary’ is ‘comparison’.

‘Conclusions’ should be, apart presented, detailed results, a bit extended by more synthetic remarks, assessing instigation features.

Reviewer 5 Report

Your work is interesting, written well, and organized.

This manuscript reports on a study of Modelling 2050 water retention scenarios as an adaptation to climate change, using the SWAT model: The case of the Bystra catchment, Poland. The study design meets the general standards and from what I can judge the data is being collected and analyzed appropriately. This work is an unpublished manuscript with relevant information that should be made public in a scientific journal for discussion among scientists working in the field.

However, there are some comments that should be considered before publishing, in this way, the social and scientific relevance of the manuscript would be improved:

Line 6: should say: changes in the soil

Line 25 should say: in the soil

Line 66: research for several

Line 79: as well as a few weather

Line 85: changing climate in Poland

Line 243: program to obtain

Line 243: picture based on the

Line 135-140: delete the paragraph.

Line 142-146: delete the paragraph

Line 187: add a table with the information of 2.3. Data used in the SWAT model

Table 1. only the first row is in bold, the rest of the table is without bold

Tables 4-16: Authors should set some value or threshold for this definition: Dark red and dark blue shading indicates large changes, while light red and light blue shading indicates small changes (author's own study).

Discussion

I continue to add a paragraph that summarizes the importance, usefulness, and social relevance, contemporary of the study, specifically pointing out the Impact, Benefit, and Projection, something like this (for example):

Line 840: These results are also evident in most tropical basins in Africa [Lagesi et al. 87] and Latin America [88, 89, 90]. It was also found [88] the strong influence that precipitation has on the hydrological response of the tropical river basin in Colombia. These results are consistent with other studies conducted under similar conditions [91, 92, 93, 94]. According to the results found, the future water supply of tropical basins will be less than the current one. These results agree with studies in tropical zones [95, 96], which argue that assuming that future water availability will be equal to the current one is unreliable in the conditions of a changing world. This statement brings at least two challenges for water managers: First, to implement methodologies for evaluating management strategies based on scenarios; and second, to propose adaptation strategies that mitigate the potential effects of climate change. In this sense, one of the advantages of hydrological models is the exploration of the consequences of different conditions and alternatives for water management.

References

In the text, reference numbers should be placed in square brackets [ ], and placed before the punctuation; for example [1], [1–3] or [1,3]. For embedded citations in the text with pagination, use both parentheses and brackets to indicate the reference number and page numbers; for example [5] (p. 10). or [6] (pp. 101–105).

The reference list should include the full title, as recommended by the ACS style guide. Style files for Endnote and Zotero are available.

References should be described as follows, depending on the type of work:

ï‚·  Journal Articles:
1. Author 1, A.B.; Author 2, C.D. Title of the article. Abbreviated Journal Name YearVolume, page range.

ï‚·  Books and Book Chapters:
2. Author 1, A.; Author 2, B. Book Title, 3rd ed.; Publisher: Publisher Location, Country, Year; pp. 154–196.
3. Author 1, A.; Author 2, B. Title of the chapter. In Book Title, 2nd ed.; Editor 1, A., Editor 2, B., Eds.; Publisher: Publisher Location, Country, Year; Volume 3, pp. 154–196.

ï‚·  Unpublished materials intended for publication:
4. Author 1, A.B.; Author 2, C. Title of Unpublished Work (optional). Correspondence Affiliation, City, State, Country. year, status (manuscript in preparationto be submitted).
5. Author 1, A.B.; Author 2, C. Title of Unpublished Work. Abbreviated Journal Name year, phrase indicating stage of publication (submittedacceptedin press).

ï‚·  Unpublished materials not intended for publication:
6. Author 1, A.B. (Affiliation, City, State, Country); Author 2, C. (Affiliation, City, State, Country). Phase describing the material, year. (phase: Personal communication; Private communication; Unpublished work; etc.)

ï‚·  Conference Proceedings:
7. Author 1, A.B.; Author 2, C.D.; Author 3, E.F. Title of Presentation. In Title of the Collected Work (if available), Proceedings of the Name of the Conference, Location of Conference, Country, Date of Conference; Editor 1, Editor 2, Eds. (if available); Publisher: City, Country, Year (if available); Abstract Number (optional), Pagination (optional).

ï‚·  Thesis:
8. Author 1, A.B. Title of Thesis. Level of Thesis, Degree-Granting University, Location of University, Date of Completion.

ï‚·  Websites:
9. Title of Site. Available online: URL (accessed on Day Month Year).
Unlike published works, websites may change over time or disappear, so we encourage you create an archive of the cited website using a service such as 
WebCite. Archived websites should be cited using the link provided as follows:
10. Title of Site. URL (archived on Day Month Year).

References

I suggest adding recent references which address the issue in question in Latin American territories. I suggest incorporating the new recommended references in the discussion section, that would improve the scientific quality of the candidate manuscript. Suggested citations are for genuine scientific reasons that emphasize the current topic of study in context:

87. Legesse, D.; Vallet-Coulomb, C.; Gasse, F. Hydrological response of a catchment to climate and land use changes in Tropical Africa: Case study South Central Ethiopia. Journal of Hydrology 2003, 275, 67–85.

88. Romero-Cuéllar, J.; Buitrago-Vargas, A.; Quintero-Ruiz, T.; Francés F. Simulación hidrológica de los impactos potenciales del cambio climático en la cuenca hidrográfica del río Aipe, en Huila, Colombia. Ribagua 2018 2;5(1), 63-78. https://doi.org/10.1080/23863781.2018.1454574

89.Parra, R.; Cortez, A.; Orlando, B. Characterization of precipitation patterns in Anzoátegui state, Venezuela. Ería 2017, 3 (3): 353-365. https://doi.org/10.17811/er.3.2017.353-365

90. Olivares, B.O.; Zingaretti, M.L. Application of multivariate methods for the characterization of periods of meteorological drought in Venezuela. Luna Azul 2019(48),172-92. https://doi.org/10.17151/luaz.2019.48.10

91. Zingaretti, M.L.; Campos, O. Analysis of the meteorological drought in four agricultural locations of Venezuela by the combination of multivariate methods. Cuad. de Invest. UNED 2018, 10(1), 192-203. http://dx.doi.org/10.22458/urj.v10i1.2026

92.Cortez, A.; Parra, R.; Lobo, D.; Rodríguez, M.F.; Rey, J.C. Descripción de los eventos de sequía meteorológica en localidades de la cordillera central, Venezuela. Ciencia, Ingenierías y Aplicaciones 2018, I (1),22-44. http://dx.doi.org/10.22206/cyap.2018.vlil.pp23-45.

93. Parra, R.M.; Cortez, A.; Lobo, D.; Rey, J.C.; Rodríguez, M.F. Characteristics of the meteorological drought (1980-2014) in two agricultural localities of the Venezuelan Andes. Rev. Invest. 2018, 42(95), 38-55.

94. Hernandez, R.; Orlando, B. Application of multivariate techniques in the agricultural land’s aptitude in Carabobo, Venezuela. Trop. Subtrop. Agroecosystems 2020, 23(2),1-12. https://n9.cl/zeedh

95. Casana, S.; Campos, O. Evolution, and trend of surface temperature and windspeed (1994 - 2014) at the Parque Nacional Doñana, Spain. Rev. Fac. Agron. (LUZ). 2020, 37(1),1-25.  https://n9.cl/c815e

96. Olivares, B.O. Tropical rainfall conditions in rainfed agriculture in Carabobo, Venezuela. La Granja 2018, 27, 86–102. https://doi.org/10.17163/lgr.n27.2018.07.

Round 2

Reviewer 1 Report

I am happy with the changes made by the authors. Some minor edits are needed to address some language issues, for example, emergency is misspelled as emeregency.

Author Response

Dear Reviewer,

thank you for your commentary on my paper. The comments were very valuable and allowed me to take a deeper look at some of the paragraphs in the paper.

With sincere greetings,

Damian Badora

Reviewer 5 Report

Después de revisar el documento pdf, no puedo ver claramente los cambios de los autores. Lo siento, pido a los autores que muestren los cambios realizados en el documento.

Author Response

Dear Reviewer,

all changes have been applied in *.docx file. Your valuable suggestions, are also included there.

Best regards,

Damian Badora

Round 3

Reviewer 5 Report

The authors made the suggested changes in the manuscript, therefore I recommend accepting for publication